# LOCAL FLOW MATCHING GENERATIVE MODELS

## ABSTRACT

Flow Matching (FM) is a simulation-free method for learning a continuous and invertible flow to interpolate between two distributions, and in particular to generate data from noise in generative modeling. In this paper, we introduce Local Flow Matching (LFM), which consecutively learns a sequence of FM sub-models and each matches a diffusion process up to the time of the step size in the data-to-noise direction. In each step, the two distributions to be interpolated by the sub-model are closer to each other than data vs. noise, and this enables the use of smaller models with faster training. The stepwise structure of LFM is natural to be distilled and different distillation techniques can be adopted to speed up generation. Theoretically, we prove a generation guarantee of the proposed flow model in terms of the $\chi^2$-divergence between the generated and true data distributions. In experiments, we demonstrate the improved training efficiency and competitive generative performance of LFM compared to FM on the unconditional generation of tabular data and image datasets, and also on the conditional generation of robotic manipulation policies.

## 1 INTRODUCTION

Generative modeling has revolutionized the field of machine learning, enabling the creation of realistic synthetic data across various domains. Recently, diffusion models (Ho et al., 2020; Song et al., 2021) started to take over earlier models like Generative Adversarial Networks (GAN) (Goodfellow et al., 2014), Variational Autoencoders (VAE) (Kingma & Welling, 2014) and Normalizing Flows (Kobyzev et al., 2020), offering benefits in terms of stability, diversity, and scalability. Diffusion models have found diverse applications in audio synthesis (Kong et al., 2021), text-to-image synthesis (Rombach et al., 2022), imitation learning for robotics (Chi et al., 2023), among others. A notable advantage of score-based diffusion models is their simulation-free training, meaning that the training objective is a "score matching" loss taking the form of an $L^2$ loss averaged over data samples. Under a Stochastic Differential Equation (SDE) formulation (Song et al., 2021), this allows training of a continuous-time score function parametrized by a neural network from a large amount of data in high dimensions.

Compared to the SDE generation in diffusion models, the Ordinary Differential Equation (ODE) generation of a trained diffusion or flow model is deterministic and typically uses fewer time steps. Using an ODE formulation, Flow Matching (FM) models (Lipman et al., 2023; Albergo & Vanden-Eijnden, 2023; Liu, 2022) proposed a simulation-free training of flow models based on regressing vector fields using an $L^2$ "matching" loss and have shown state-of-the-art generation performance on various tasks, including text-to-image generation (Esser et al., 2024), humanoid robot control (Rouxel et al., 2024), audio generation (Guan et al., 2024), and application to discrete data in programming code generation (Gat et al., 2024). The simulation-free training of flow models significantly alleviates the computational issue of earlier Continuous Normalizing Flow (CNF) models using likelihood-based training Grathwohl et al. (2018). Going beyond the fast generation of ODE flow models, recent works on model distillation (Salimans & Ho, 2022; Liu et al., 2023; Song et al., 2023) have shown that the generation of large (diffusion and flow) generative models can be significantly accelerated and in some cases computed in an extremely small number of steps.

In this work, we propose a simulation-free training technique of CNFs called "Local Flow Matching" (LFM), which can be seen as a stepwise version of FM, and we call previous FMs the *global* ones. The proposed LFM trains a sequence of small (sub-)flow models, which, after concatenation, transport invertibly from data to noise and back. In each sub-flow, any FM model can be plugged

in. In other words, the global FM tries to interpolate directly between noise and data distributions which may differ a lot, while our local FM breaks down the job into smaller steps and interpolates between a pair of distributions which are closer to each other (namely "local") in each step, see Figure 1. Because the source and target distributions are not too far away, we expect to train a local FM model with potentially smaller model and faster convergence in training. This would allow reduced memory load and computational cost without sacrificing model quality. In addition to the benefits when training from scratch, our framework is compatible with various distillation techniques, and we empirically found that our model can have advantage over global FMs after distillation.

Specifically, in each step of LFM, we train a sub-flow model to interpolate between $(p_{n-1}, p_n^*)$, where $p_n^*$ evolves from $p_{n-1}$ along the diffusion process for time up to the step size. The forward process (data-to-noise) starts from $p_0$ being the data distribution and ends at some $p_N$ which is close to normal distribution. The reverse process (noise-to-data) uses the invertibility of each sub-flow model to generate data from noise by ODE integration. The construction of using (the marginal distribution of) a diffusion process as a "target" in each step allows us to theoretically prove the generation guarantee of LFM by connecting to the diffusion theory.

The stepwise approach in our work is inspired by similar constructions in the literature, including block-wise training of ResNet under the GAN framework Johnson & Zhang (2019) and flow-based generative models implementing a discrete-time gradient descent in Wasserstein space following the Jordan-Kinderleherer-Otto (JKO) scheme (Alvarez-Melis et al., 2022; Mokrov et al., 2021; Xu et al., 2023b; Vidal et al., 2023). Unlike previous models, the proposed LFM is simulation-free, which allows application to high-dimensional data like large images and robotics data as shown in our experiments.

In summary, the contributions of the work are:

- To generate data from the normal distribution, we propose to train a consecutive series of FM sub-models which approximately follow the diffusion process in the data-to-noise direction. Such breakdown of the global flow into small pieces makes the pair of distributions closer to each other in each step, and then the FM sub-model can be obtained with reduced model size and faster training convergence. Concatenation of all the sub-models provides an invertible flow that can go from noise to data in the reverse direction.

- Theoretically, we prove the generation guarantee, namely how the generated distribution by LFM approximates the data distribution $P$, under $\chi^2$ divergence which implies Kullback–Leibler (KL) and Total Variation (TV) guarantees. We prove an $O(\varepsilon^{1/2})$-$\chi^2$ guarantee, where $\varepsilon$ is the $L^2$ error of FM training objective, and the other technical assumptions are motivated by our stepwise FM to the OU process. Our theory applies when data density is regular and also covers cases when $P$ merely has finite second moments (where we use a short-time initial diffusion to smooth $P$), e.g., when $P$ is compactly supported.

- Our framework allows readily plugging in different designs of the FM in each local sub-flow model. In addition, the stepwise structure of LFM renders it natural to be distilled, and our approach is compatible with different distillation techniques. Empirically, the proposed LFM shows improved training efficiency and competitive generative performance against existing FM methods on likelihood estimation, image generation, and robotic manipulation tasks. The model gives strong performance when trained from scratch and in the distilled setting.

**Notations.** We use the same notation for the distribution and its density (with respect to the Lebesgue measure on $\mathbb{R}^d$) when there is no confusion. For a distribution $P$, $M_2(P) := \int_{\mathbb{R}^d} \|x\|^2 dP(x)$. Let $\mathcal{P}_2 = \{P \text{ on } \mathbb{R}^d, s.t., M_2(P) < \infty\}$. The Wasserstein-2 distance, denoted by $\mathcal{W}_2(P, Q)$, gives a metric on $\mathcal{P}_2$. For $T : \mathbb{R}^d \to \mathbb{R}^d$, $T_\# P$ denotes the *pushforward* of $P$, i.e., $T_\# P(A) = P(T^{-1}(A))$ for a measurable set $A$. We also write $T_\# p$ for the pushforwarded density.

## 1.1 RELATED WORKS

**Continuous normalizing flow (CNF).** CNF uses a neural ODE model (Chen et al., 2018) optimized by maximizing the model likelihood, which the ODE parametrization can compute, on observed data samples (Grathwohl et al., 2018). To facilitate training and inference, subsequent works

have proposed advanced techniques such as trajectory regularization (Finlay et al., 2020; Onken et al., 2021) and block-wise training (Fan et al., 2022; Xu et al., 2023b). These techniques help stabilize the training process and improve the model's performance. Despite successful applications in time-series analyses (de Bézenac et al., 2020), uncertainty estimation (Berry & Meger, 2023), optimal transport (Xu et al., 2023a), and astrophysics (Langendorff et al., 2023), a main drawback of CNF is its computational cost, since backpropagating the neural ODE in likelihood-based training is expensive (non-simulation free) and not scalable to high dimensions.

**Simulation-free flow models.**    Flow Matching (FM) models (Lipman et al., 2023; Albergo & Vanden-Eijnden, 2023; Liu, 2022) are simulation-free and a leading class of generative models. We review the technical details of FM in Section 3.1. FM methods are compatible with different choices to interpolate two random end-points drawn from the source and target distributions, e.g. straight lines (called "Optimal Transport (OT) path") or motivated by the diffusion process (called "diffusion path") (Lipman et al., 2023). Later works also considered pre-computed OT interpolation (Tong et al., 2023), and stochastic interpolation paths (Albergo et al., 2023). All previous works train a *global* flow model to match between the two distributions, which could require a large model that takes a longer time to train. In this work, we propose to train multiple smaller flow models. Our approach is compatible with any existing FM method to train these so-called local flows, making it a flexible and extensible framework.

**Accelerated generation and model distillation.**    Model compression and distillation have been intensively developed to accelerate the generation of large generative models. Baranchuk et al. (2021) proposed learning a compressed student normalizing flow model by minimizing the reconstruction loss from a teacher model. For diffusion models, progressive distillation was developed in (Salimans & Ho, 2022), and Consistency Models (Song et al., 2023) demonstrated high quality sample generation by directly mapping noise to data. For FM models, (Liu et al., 2023) proposed to distill the ODE trajectory into a single mapping parametrized by a network, which can reduce the number of function evaluations (NFE) to be one. The approach was later effectively applied to large text-to-image generation (Liu et al., 2024). More recent techniques to distill FM models include dynamic programming to optimize stepsize given a budget of NFE (Nguyen et al., 2024). In our work, each local flow model can be distilled into a single-step mapping following (Liu et al., 2023), and the model can be further compressed if needed. Our framework is compatible with different distillation techniques.

**Theoretical guarantees of generative models.**    Guarantees of diffusion models, where the generation process utilizes an SDE (random) Lee et al. (2023); Chen et al. (2022; 2023a); Benton et al. (2024b) or ODE (deterministic) sampler Chen et al. (2023b; 2024); Li et al. (2024a;b); Huang et al. (2024), have been recently intensively developed. In comparison, there are fewer theoretical findings for ODE flow models both in training (forward process) and generation (reverse processes) like CNF or FM. For FM models, $\mathcal{W}_2$-guarantee was proved in Benton et al. (2024a) and in (Gao et al., 2024) with sample complexity analysis. The Wasserstein bound does not imply KL or TV bound, which are more relevant for information-theoretical and statistical interpretation. For CNF trained by maximizing likelihood, non-parametric statistical convergence rates were proved in (Marzouk et al., 2024). KL guarantee of a step-wisely trained CNF model motivated by JKO scheme (Xu et al., 2023b) was proved in (Cheng et al., 2024), yet the approach is likelihood-based and not simulation-free. Recently, (Silveri et al., 2024) proved KL guarantee for an SDE interpolation version of the FM model introduced in (Albergo et al., 2023). Our work analyzes a stepwise ODE FM model, which is simulation-free, and the guarantee is in $\chi^2$ divergence which implies KL (and TV) guarantees.

## 2    PRELIMINARIES

**Flow models and neural ODE.**    In the context of generative models, the goal is to generate data distribution $P$, which is usually only accessible from finite training samples. When $P$ has density we denote it by $p$. A continuous-time flow model trains a neural ODE (Chen et al., 2018) to generate data distribution $p$ from a standard distribution $q$, which is typically $\mathcal{N}(0, I)$ and called "noise".

Specifically, a neural ODE model provides a velocity field $v(x, t; \theta)$ on $\mathbb{R}^d \times [0, T]$ parametrized by a neural network, and $\theta$ consists of trainable parameters. In a flow model, the solution of the ODE

$$\dot{x}(t) = v(x(t), t; \theta), \quad t \in [0, T], \quad x(0) \sim \rho_0, \tag{1}$$

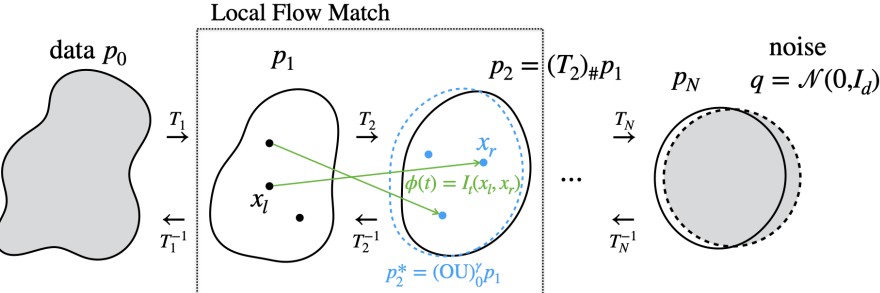

Figure 1: Illustration of the proposed LFM model. In the $n$-step step, a local FM model is trained to interpolate from $p_{n-1}$ to $p_n^*$ and it gives the learned (invertible) transport map $T_n$ which pushforwards $p_{n-1}$ to $p_n$. The concatenation of the $N$ sub-models gives a flow between data and noise.

where $\rho_0$ is a distribution, and we denote the law of $x(t)$ as $\rho_t$. Whenever the ODE is well-posed, it gives a continuous invertible mapping from the initial value $x(0)$ to the terminal value $x(T)$, and the inverse map from $x(T)$ to $x(0)$ can be computed by integrating the ODE (1) reverse in time. Thus, we can either set $\rho_0$ to be noise and $\rho_T$ to be data or the other way round. Earlier flow models like (continuous-time) CNFs (Grathwohl et al., 2018) train the flow $v(x, t; \theta)$ based on likelihood, i.e. letting $\rho_0$ be noise and maximizing the likelihood of $\rho_T$ on data samples. Such approaches are not simulation-free and can be difficult to scale to large-size computations.

**Ornstein-Uhlenbeck (OU) process.** The OU process in $\mathbb{R}^d$ observes the following SDE

$$dX_t = -X_t dt + \sqrt{2}dW_t, \tag{2}$$

where the equilibrium density is $q \propto e^{-V}$ with $V(x) = \|x\|^2/2$. In other words, $q$ is the standard normal density $\mathcal{N}(0, I)$. Suppose $X_0 \sim \rho_0$ which is some initial density at time zero (the initial distribution may not have density), and denote by $\rho_t$ the marginal density of $X_t$ for $t > 0$. The time evolution of $\rho_t$ is described by the Fokker–Planck Equation (FPE) $\partial_t \rho_t = \nabla \cdot (\rho_t \nabla V + \nabla \rho_t)$, $V(x) = \|x\|^2/2$, which determines $\rho_t$ given the initial value $\rho_0$. We also introduce the operator $(\text{OU})_0^t$ and write

$$\rho_t = (\text{OU})_0^t \rho_0. \tag{3}$$

Equivalently, $\rho_t$ is the probability density of the random vector $Z_t := e^{-t}X_0 + \sigma_t Z$, where $\sigma_t^2 := 1 - e^{-2t}$, $Z \sim \mathcal{N}(0, I_d)$ and is independent from $X_0$.

## 3 LOCAL FLOW MATCHING (LFM)

The essence of the proposed LFM model is to break down a single flow from data to noise (and back) into several pieces, and apply FM on each piece sequentially. We introduce the method in this section, and further algorithmic details are provided in Section 5.

### 3.1 REVIEW OF FLOW MATCHING

Flow Matching (Lipman et al., 2023), also proposed as "stochastic interpolants" (Albergo & Vanden-Eijnden, 2023) and "rectified flow" (Liu, 2022), trains the flow $v(x, t; \theta)$ by minimizing an $L^2$ loss and is simulation-free; our stepwise approach in this work will be based on FM. Rescale the time interval to be $[0, 1]$. FM utilizes a pre-specified interpolation function $I_t$ given two endpoints $x_l$ and $x_r$ ($_l$ for 'left' and $_r$ for 'right') as

$$\phi(t) := I_t(x_l, x_r), \quad t \in [0, 1], \tag{4}$$

where $x_l \sim p$, $x_r \sim q$, and $I_t$ can be analytically designed. The (population) training objective is

$$\min_\theta \int_0^1 \mathbb{E}_{x_l, x_r} \|v(\phi(t), t; \theta) - \frac{d}{dt}\phi(t)\|_2^2 dt, \tag{5}$$

and, as has been proven in the literature, the velocity field $v$ that minimizes (5) has the expression $v(x, t) = \mathbb{E}_{x_l, x_r}[\frac{d}{dt}I_t(x_l, x_r)|I_t(x_l, x_r) = x]$ and it induces a flow that transports from $p$ to $q$. We call this $v$ the *target* velocity field, and say that the FM model interpolates the pair of distributions $(p, q)$. See Section 5.1 for more algorithmic details of FM, such as the choice of $I_t$.

## 3.2 STEPWISE TRAINING OF $N$ FM SUB-MODELS

We propose to train a sequence of $N$ FM sub-models (called sub-flows) on time $[0, T]$, each by its own training objective to interpolate a pair of distributions, and collectively the $N$ sub-flows fulfill the transport from data $p$ to noise $q$ (and back by the reverse flow). Our method will train the $N$ sub-flows sequentially, where in each step the flow tries to match the terminal density evolved by the OU process of time up to the step size. Because the step size is not large and the pair of end-point distributions to interpolate are not far away, we call our method *local* FM.

**Training.** On the time interval $[0, T]$, we first specify a time step schedule $\{\gamma_n\}_{n=1}^N$, $\sum_n \gamma_n = T$, and for the simplest case $\gamma_n = \gamma = T/N$. Suppose the data distribution $P$ has a regular density $p_0$ – if not, we can apply a short-time diffusion to $P$, and take the resulting density as $p_0$, see more in Section 4.3. Starting from $p_0$ (when $n = 1$), we recursively construct target density $p_n^*$ by

$$p_n^* = (\text{OU})_0^{\gamma_n} p_{n-1},$$

where the operator $(\text{OU})_0^t$ is as defined in (3). In other words, let $x_l \sim p_{n-1}$, and

$$x_r := e^{-\gamma_n} x_l' + \sqrt{1 - e^{-2\gamma_n}} g, \quad x_l' \sim p_{n-1}, \quad g \sim \mathcal{N}(0, I_d), \quad g \perp x_l', \tag{6}$$

then the marginal distribution of $x_r$ is $p_n^*$. We now train the $n$-th sub-flow, denoted by $\hat{v}_n(x, t; \theta)$, by FM to interpolate the pair $(p_{n-1}, p_n^*)$ using (4)(5) (the time interval $[0, \gamma_n]$ is rescaled to be $[0, 1]$ in FM training). The velocity field $\hat{v}_n$ can have its own parametrization $\theta_n$, which then can be trained independently from previous sub-flows.

After the sub-flow $\hat{v}_n$ is trained, it provides a transport map $T_n$ that maps $x_{n-1} \sim p_{n-1}$ to $x_n$ by

$$x_n := T_n(x_{n-1}) = x(\gamma_n) = x_{n-1} + \int_0^{\gamma_n} \hat{v}_n(x(t), t; \theta) dt, \tag{7}$$

where $x(t)$ solves the ODE $\dot{x}(t) = \hat{v}_n(x(t), t; \theta)$ from $x(0) = x_{n-1}$. The mapping $T_n$ is invertible and $T_n^{-1}$ is by integrating the reverse-time ODE. We denote the distribution of $x_n$ as $p_n$, that is,

$$p_n = (T_n)_\# p_{n-1}.$$

From this $p_n$, we can train the next sub-flow by interpolate $(p_n, p_{n+1}^*)$. This recursive scheme is illustrated in Figure 1.

If the flow matching in $n$-th step is successful, then we expect the trained $\hat{v}_n$ makes $p_n \approx p_n^*$. Define the time stamps $\{t_n\}_{n=0}^N$, $t_0 = 0$, $t_n - t_{n-1} = \gamma_n$, and then $t_N = T$. If $p_n$ exactly equals $p_n^* = (\text{OU})_0^{\gamma_n} p_{n-1}$, then over $N$ steps we have $p_N = (\text{OU})_0^T p_0$, which approximates the equilibrium $q$ exponentially fast as $T$ increases. In practice, the trained flow has some finite flow matching error and $p_n$ has some discrepancy from $p_n^*$, yet if the error is small, we still expect $p_N \approx q$, assuming $T = \sum_n \gamma_n$ is sufficiently large. This will be theoretically analyzed in Section 4.

In practice, when training the $n$-step sub-flow, the finite samples of $p_{n-1}$ are obtained by transporting samples from $p_0$ through the previous sub-flows. This pushforward from $p_{n-1}$ to $p_n$ can be done for all training samples once the $n$-th sub-flow is trained, see Algorithm 1. The proposed LFM model can be trained from scratch, and we also distill the model to improve generation efficiency. See Section 5 for details.

**Generation.** Once the $N$ sub-flows are trained, we go backward from the $N$-th to the 1st sub-flows to generate data from noise. Specifically, we sample $y_N \sim q$, and let $y_{n-1} = T_n^{-1}(y_n)$ for $n = N, \cdots, 1$, where $T_n^{-1}$ is computed by integrating the ODE with velocity field $\hat{v}_n$ reverse in time. We then use $y_0$ as the generated data samples. The closeness of the distribution of $y_0$ to the data distribution $P$ will be theoretically shown in Section 4.

## 4 THEORETICAL GUARANTEE OF GENERATION

In this section, we prove the generation guarantee, namely how the generated density by the trained LFM model is close to the true data distribution. All proofs are left to Appendix A.

### 4.1 SUMMARY OF FORWARD AND REVERSE PROCESSES

Recall that $P$ is the distribution of data in $\mathbb{R}^d$, and $q$ the density of $\mathcal{N}(0, I_d)$. The procedures of training and generation in Section 3 can be summarized in the following forward (data-to-noise training) and reverse (noise-to-data generation) processes respectively:

$$
\begin{aligned}
\text{(forward)} \quad & p = p_0 \xrightarrow{T_1} p_1 \xrightarrow{T_2} \cdots \xrightarrow{T_N} p_N \approx q, \\
\text{(reverse)} \quad & p \approx q_0 \xleftarrow{T_1^{-1}} q_1 \xleftarrow{T_2^{-1}} \cdots \xleftarrow{T_N^{-1}} q_N = q,
\end{aligned}
\tag{8}
$$

where $T_n$ is by the learned $n$-th step sub-flow as defined in (7). When $P$ has a regular density $p$ we set it as $p_0$; otherwise, $p_0$ will be a smoothed version of $P$, see more below. The reverse process gives the final output samples which have density $q_0$. The goal of our analysis is to show that $q_0 \approx p$, namely the generated density is close to the data density.

To keep exhibition simple, we consider when $\gamma_n = \gamma > 0$ for all $n$. Using the time stamps $t_n$, the $n$-th step sub-flow is on the time interval $[t_{n-1}, t_n]$, which can be shifted to be $[0, \gamma]$. Our analysis will be based on comparing the true or target flow (that transports to terminal density $p_n^*$) with the learned flow (that transports to terminal density $p_n$), and we introduce the notations for the corresponding transport equations.

For fixed $n$, on the shifted time interval $[0, \gamma]$, the target flow and the learned flow are induced by the velocity field $v(x, t)$ and $\hat{v}(x, t) = \hat{v}_n(x, t; \theta)$ respectively. We omit $_n$ in the notation. As was shown in Section 3.1, the target $v$ depends on the choice of $I_t$ yet it always transports from $p_{n-1}$ to $p_n^*$. Let $\rho_t(x, t)$ be the law of $x(t)$ that solves the ODE with velocity field $v$, where $x(0) \sim p_{n-1}$, and $\hat{\rho}_t(x, t)$ be the law of $x(t)$ that solves the ODE with the learned $\hat{v}$. ($\rho_t$ is not necessarily the density of an OU process $X_t$, though $\rho_\gamma = p_n^* = (\text{OU})_0^{\gamma_n} p_{n-1}$.) We also denote $\rho(x, t)$ as $\rho_t$ (omitting the variable $x$) or $\rho$ (omitting both $x$ and $t$), depending on the context, and similarly for $\hat{\rho}(x, t)$. The target and learned flows have the transport equations as

$$
\begin{aligned}
\partial_t \rho + \nabla \cdot (\rho v) = 0, \quad & \rho_0 = p_{n-1}, \quad \rho_\gamma = p_n^*. \\
\partial_t \hat{\rho} + \nabla \cdot (\hat{\rho} \hat{v}) = 0, \quad & \hat{\rho}_0 = p_{n-1}, \quad \hat{\rho}_\gamma = p_n.
\end{aligned}
\tag{9}
$$

### 4.2 EXPONENTIAL CONVERGENCE OF THE FORWARD PROCESS IN $\chi^2$

We introduce the following assumption about the learned flow $\hat{v}_n$:

**Assumption 1.** *(A0) For all $n$, the learned $\hat{v}_n$ ensures that $T_n$ and $T_n^{-1}$ are non-degenerate (Definition A.1), and, on the time interval $[t_{n-1}, t_n]$ shifted to be $[0, \gamma]$, $\int_0^\gamma \int_{\mathbb{R}^d} \|v - \hat{v}\|^2 \rho \, dx \, dt \leq \varepsilon^2$. Without loss of generality, assume $\varepsilon < 1$.*

The training objective of FM is equivalent to minimizing the $L^2$ loss $\int_0^\gamma \int_{\mathbb{R}^d} \|v - \hat{v}\|^2 \rho \, dx \, dt$ (Albergo & Vanden-Eijnden, 2023), thus our $\varepsilon$ is the learning error of FM (in each step and uniform for all $n$).

**Assumption 2.** *There are positive constants $C_1, C_2, L$ such that, for all $n$, on the time interval $[t_{n-1}, t_n]$ shifted to be $[0, \gamma]$,*

*(A1) $\rho_t, \hat{\rho}_t$ for any $t \in [0, \gamma]$ are positive on $\mathbb{R}^d$ and $\rho_t(x), \hat{\rho}_t(x) \leq C_1 e^{-\|x\|^2/2}$;*

*(A2) $\forall t \in [0, \gamma]$, $\rho_t, \hat{\rho}_t$ are $C^1$ on $\mathbb{R}^d$ and $\|\nabla \log \rho_t(x)\|, \|\nabla \log \hat{\rho}_t(x)\| \leq L(1 + \|x\|), \forall x \in \mathbb{R}^d$;*

*(A3) $\forall t \in [0, \gamma]$, $\int_{\mathbb{R}^d} (1 + \|x\|)^2 (\rho_t^3/\hat{\rho}_t^2)(x) dx \leq C_2$.*

At $t = 0$, $\rho_0 = \hat{\rho}_0 = p_{n-1}$, thus Assumption 2 requires that $f = p_n$ for $n = 0, 1, \cdots$ satisfies

$$
f(x) \leq C_1 e^{-\|x\|^2/2}, \quad \|\nabla \log f(x)\| \leq L(1 + \|x\|), \quad \int_{\mathbb{R}^d} (1 + \|x\|)^2 f(x) dx \leq C_2, \tag{10}
$$

by (A1)(A2)(A3) respectively. The first two inequalities require $p_n$ to have a Gaussian decay envelope and the score of $p_n$ has linear growth (can be induced by Lipschitz regularity), and the third inequality can be implied by the first one. The condition (10) poses regularity conditions on $p_0$ which can be satisfied by many data densities in applications, and, in particular, if $P$ has finite support then these hold after $P$ is smoothed by an initial short-time diffusion (Lemma A.6).

Technically, Assumption 2 poses the Gaussian envelope and regularity requirements on all $p_n$ and also $\rho_t$ and $\hat{\rho}_t$ for all time. This can be expected to hold at least when FM is well-trained: suppose $\rho_t$

satisfies (A1)(A2) for all $t$ due to the regularity of $v$ (by the analytic $I_t$ and the regularity of the pair of densities $(p_{n-1}, p_n^*)$), when the true and learned flows match each other, we have $\hat{\rho} \approx \rho$, thus we also expect (A1)(A2) to hold for $\hat{\rho}_t$; Meanwhile, the ratio $\rho_t/\hat{\rho}_t$ is close to 1 and if can be assumed to be uniformly bounded, then (A3) can be implied by the boundedness of $\int_{\mathbb{R}^d} (1 + \|x\|)^2 \rho_t(x) dx$ which can be implied by the Gaussian envelope (A1) of $\rho_t$.

**Proposition 4.1** (Exponential convergence of the forward process). *Under Assumptions 1-2,*

$$\chi^2(p_n\|q) \le e^{-2\gamma n}\chi^2(p_0\|q) + \frac{C_4}{1-e^{-2\gamma}}\varepsilon^{1/2} \tag{11}$$

*for $n = 1, 2, \cdots$, where the constant $C_4$ defined in* (20) *is determined by $C_1$, $C_2$, $L$, $\gamma$ and $d$.*

### 4.3 GENERATION GUARANTEE OF THE BACKWARD PROCESS

$P$ **with regular density.** Because the composed transform $T_N \circ \cdots \circ T_1$ from $p_0$ to $p_N$ is invertible, and the inverse map transforms from $q = q_N$ to $q_0$, the smallness of $\chi^2(p_N\|q_N)$ implies the smallness of $\chi^2(p_0\|q_0)$ due to a bi-directional version of data processing inequality (DPI), see Lemma A.4, As a result, the exponential convergence of the forward process in Proposition 4.1 directly gives the $\chi^2$-guarantee $q_0 \approx p$.

**Theorem 4.2** (Generation guarantee of regular data density). *Suppose $P \in \mathcal{P}_2$ has density $p$, let $p_0 = p$ and $p_0$ satisfies* (10). *Under Assumptions 1-2, if we use the number of steps $N \ge \frac{1}{2\gamma}\left(\log \chi^2(p_0\|q) + \frac{1}{2}\log(\frac{1}{\varepsilon})\right) \sim \log(1/\varepsilon)$, we have $\chi^2(p\|q_0) \le C\varepsilon^{1/2}$, $C := (1 + \frac{C_4}{1-e^{-2\gamma}})$.*

By that $\mathrm{KL}(p\|q) \le \chi^2(p\|q)$ (Lemma A.5), the $\chi^2$-guarantee of $O(\varepsilon^{1/2})$ in Theorem 4.2 implies that $\mathrm{KL}(p\|q_0) = O(\varepsilon^{1/2})$, and consequently $\mathrm{TV}(p, q_0) = O(\varepsilon^{1/4})$ by Pinsker's inequality.

$P$ **up to initial diffusion.** For data distribution $P$ that may not have a density or the density does not satisfy the regularity conditions, we introduce a short-time diffusion to obtain a smooth density $\rho_\delta$ from $P$ and use it as $p_0$. This construction can ensure that $\rho_\delta$ is close to $P$, e.g. in $\mathcal{W}_2$ distance, and is commonly used in diffusion models (Song et al., 2021) and also the theoretical analysis (known as "early stopping") (Chen et al., 2023a).

**Corollary 4.3** (Generation of $P$ up to initial diffusion). *Suppose $P \in \mathcal{P}_2$ and for some $\delta < 1$, $p_0 = \rho_\delta = (\mathrm{OU})_0^\delta(P)$ satisfies* (10) *for some $C_1$, $C_2$ and $L$. With $p_0 = \rho_\delta$, suppose Assumptions 1-2 hold, then for $N$ and $C$ are as in Theorem 4.2, the generated density $q_0$ of the reverse process makes $\chi^2(\rho_\delta\|q_0) \le C\varepsilon^{1/2}$. Meanwhile, $\mathcal{W}_2(P, \rho_\delta) \le C_5\delta^{1/2}$, $C_5 := (M_2(P) + 2d)^{1/2}$.*

In particular, as shown in Lemma A.6, when $P$ is compactly supported (not necessarily having density), then for any $\delta > 0$, there are $C_1$, $C_2$ and $L$ such that $p_0 = \rho_\delta$ satisfies (10). This will allow us to make $\mathcal{W}_2(P, \rho_\delta)$ arbitrarily small. Generally, the theoretical constants $C_1$, $C_2$ and $L$ may depend on $\delta$, and consequently so does the constant $C$ in the $\chi^2$ bound. At last, similarly to the comment beneath Theorem 4.2, the $O(\varepsilon^{1/2})$-$\chi^2$ guarantee in Corollary 4.3 of $q_0 \approx p_\delta$ implies $O(\varepsilon^{1/2})$-KL and $O(\varepsilon^{1/4})$-TV guarantee.

## 5 ALGORITHM

### 5.1 TRAINING LFM FROM SCRATCH

The training of LFM is summarized in Algorithm 1. We call each sub-flow FM a step or a "block". In each block, any FM algorithm can be adopted. In our experiments, we consider the following choices of $I_t(x_l, x_r)$ in (4), following (Lipman et al., 2023; Albergo & Vanden-Eijnden, 2023): (i) Optimal Transport: $I_t(x_l, x_r) = x_l + t(x_r - x_l)$, (ii) Trigono-metric: $I_t(x_l, x_r) = \cos(\frac{1}{2}\pi t)x_l + \sin(\frac{1}{2}\pi t)x_r$. The selection of time stamps $\{\gamma_n\}_{n=1}^{N-1}$ depends on the task and is explained in Appendix B.1. In our experiments, we use $N$ up to 10. In each sub-flow,

---

**Algorithm 1** Local Flow Matching (LFM) from scratch

**Input:** Data samples $\sim p_0$, timesteps $\{\gamma_n\}_{n=1}^{N-1}$.
**Output:** $N$ sub-flows $\{\hat{v}_n\}_{n=1}^N$
1: **for** $n = 1, \ldots, N$ **do**
2:      Draw samples $x_l \sim p_{n-1}$ and $x_r \sim p_n^* = (\mathrm{OU})_0^{\gamma_n} p_{n-1}$ by (6) (when $n = N$, let $p_n^* = q$)
3:      Innerloop FM: optimize $\hat{v}_n(x, t; \theta)$ by minimizing (5) with mini-batches
4:      **if** $n \le N - 1$ **then**
5:          Push-forward the samples $\sim p_{n-1}$ to be samples $\sim p_n$ by $T_n$ in (7)
6:      **end if**
7: **end for**

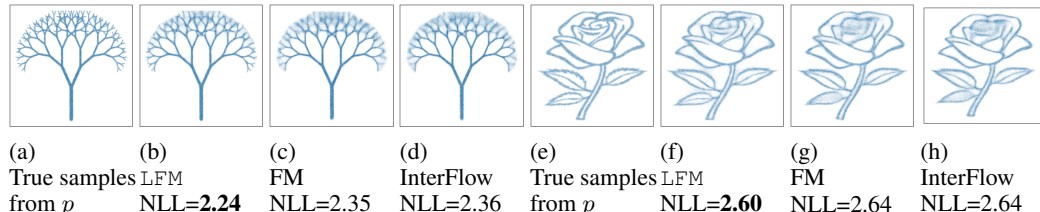

| (a) | (b) | (c) | (d) | (e) | (f) | (g) | (h) |
|-----|-----|-----|-----|-----|-----|-----|-----|
| True samples from $p$ | LFM NLL=**2.24** | FM NLL=2.35 | InterFlow NLL=2.36 | True samples from $p$ | LFM NLL=**2.60** | FM NLL=2.64 | InterFlow NLL=2.64 |

Figure 2: Generative performance and NLL comparison (lower is better) on 2D data.

there are no restrictions on the architecture of $\hat{v}_n(x, t; \theta)$. We use fully connected networks for vector data and UNets for images. When the sub-flows are independently parametrized, we reduce the model size of each block assuming that the target flow to match is simpler than a global flow, which also reduces memory load and facilitates the innerloop training of FM. In computing the pushforward (Line 5 in Algorithm 1) and in generation, the numerical integration of neural ODE follows the procedure in (Chen et al., 2018).

## 5.2 DISTILLATION OF LFM

Inspired by (Liu et al., 2023), we propose to distill an $N$-block pre-trained LFM model into $N' < N$ steps distilled model, where $N = N'k$ for some integer $k$. Each of the $N'$ sub-models can be distilled independently if parametrized independently. The detail is in Algorithm A.1. If the pre-trained LFM has independently parametrized blocks, we keep the sub-model size if $k = 1$ and increase the model size if the distillation combines original blocks ($k > 1$). The composition of the $N'$ distilled sub-models generates data from noise in $N'$ steps.

## 6 EXPERIMENTS

We apply the proposed LFM to simulated and real datasets, including tabular data (Section 6.2), image generation (Section 6.3), and robotic manipulation (Section 6.4). We demonstrate the improved training efficiency and generative performance of LFM compared to (global) flow models, and we also show the advantage of LFM after distillation on image generation.

### 6.1 TWO-DIMENSIONAL TOY DATA

The task is to generate distributions in 2D that have no analytic form, and we consider the "tree" and "rose" examples (Figure 2). We compare our LFM against previous FM models: FM (Lipman et al., 2023, Example II) and InterFlow (Albergo & Vanden-Eijnden, 2023). To ensure a fair comparison, we maintain an identical training scheme for all methods, including the specification of optimizers, batch size, and number of training batches. Further details regarding the experimental setup are provided in Appendix C. The accuracy of the trained models is evaluated using the negative log-likelihood (NLL) metric in (26). Figure 2 shows that LFM achieves good generation of the distribution and slightly better NLL in comparison.

### 6.2 TABULAR DATA GENERATION

We apply LFM to a set of tabular datasets (Papamakarios et al., 2017), where the generation performance is quantitatively measured by test NLL. We use the following baselines, including both dis-

Table 1: Test NLL (lower is better) on tabular data, where $d$ refers to the data dimension. The lowest NLL is in **bold** and the 2nd lowest NLL is underlined. All baseline values are quoted from their original publications, where "-" entries indicate this dataset was not used.

| | LFM | InterFlow | JKO-iFlow | AdaCat | OT-Flow | nMDMA | CPF | BNAF | FFJORD |
|---|---|---|---|---|---|---|---|---|---|
| POWER ($d = 6$) | -0.67 | -0.57 | -0.40 | -0.56 | -0.30 | **-1.78** | -0.52 | -0.61 | -0.46 |
| GAS ($d = 8$) | **-12.43** | -12.35 | -9.43 | -11.27 | -9.20 | -8.43 | -10.36 | -12.06 | -8.59 |
| MINIBOONE ($d = 43$) | 9.95 | 10.42 | 10.55 | 14.14 | 10.55 | 18.60 | 10.58 | **8.95** | 10.43 |
| BSDS300 ($d = 63$) | **-157.80** | -156.22 | -157.75 | - | -154.20 | - | -154.99 | -157.36 | -157.40 |

Table 2: FID (lower is better) comparison of LFM against InterFlow and FM under same model sizes. FIDs with the symbol * are quoted from the original publication. Note that InterFlow uses the Trig interpolant and FM uses the OT interpolant to train the global flow.

| | CIFAR-10 | | | Imagenet-32 | | | Flowers-128 | | |
|---|---|---|---|---|---|---|---|---|---|
| | FID | Batch size | # of batches | FID | Batch size | # of batches | FID | Batch size | # of batches |
| LFM (Trig interpolant) | **8.45** | **200** | $5 \times 10^4$ | **7.00** | 256 | $2 \times 10^5$ | **59.7** | 40 | $4 \times 10^4$ |
| InterFlow | 10.27* | 400 | $5 \times 10^5$ | 8.49* | 512 | $6 \times 10^5$ | 65.9 | 40 | $4 \times 10^4$ |
| LFM (OT interpolant) | **8.55** | 200 | $5 \times 10^4$ | **7.20** | 256 | $2 \times 10^5$ | **55.7** | 40 | $4 \times 10^4$ |
| FM | 12.30 | 200 | $5 \times 10^4$ | 7.51 | 256 | $2 \times 10^5$ | 70.8 | 40 | $4 \times 10^4$ |

crete and continuous flows, to compare against: InterFlow (Albergo & Vanden-Eijnden, 2023), JKO-iFlow (Xu et al., 2023b), AdaCat (Li et al., 2022), OT-Flow (Onken et al., 2021), nMDMA (Gilboa et al., 2021), CPF (Huang et al., 2021), BNAF (De Cao et al., 2020), and FFJORD (Grathwohl et al., 2018). Additional experimental details can be found in Appendix C. The results are shown in Table 1, where the proposed LFM is among the two best-performing methods on all datasets.

## 6.3 IMAGE GENERATION

We apply LFM to unconditional image generation of $32 \times 32$ and $128 \times 128$ images. We compare LFM with InterFlow and FM in terms of Frechet Inception Distance (FID) before and after distillation. To ensure a fair comparison, we maintain the same network size for both methods, and more details of the experimental setup are provided in Appendix C.

$32 \times 32$ **images.** We use the CIFAR-10 (Krizhevsky & Hinton, 2009) and Imagenet-32 (Deng et al., 2009) datasets. As shown in Table 2, LFM requires significantly less computation in training than InterFlow, and it achieves lower FID values. LFM also shows improved training efficiency against FM. We present generated images by LFM in Figure 3 and additionally, by the distilled LFM in Figure A.2 (NFE = 5) which shows an almost negligible reduction in visual quality.

$128 \times 128$ **images.** We use the Oxford Flowers (Nilsback & Zisserman, 2008) dataset. We first train LFM and InterFlow/FM to reach the same FIDs on the test set, with the global flows requiring 1.25-1.5x more training batches to do so. Subsequently, we distill LFM using Algorithm A.1 and InterFlow/FM according to (Liu et al., 2023). Quantitatively, Table 2 shows lower FID under the same number of training batches and Table 3 shows lower FID by LFM under 4 or 2 NFEs after distillation. This shows that in addition to efficiency in training from scratch, LFM after distillation also achieves lower FID. To show the qualitative results, we give generated images in Figure 3 and noise-to-image trajectories in Figure A.1. Figure A.3 further showcases high-fidelity images generated by LFM after distillation with 4 NFEs.

Table 3: FID comparison of LFM and InterFlow before and after distillation on Flowers $128 \times 128$.

| | Pre-distillation | Distilled @ 4 NFEs | Distilled @ 2 NFEs |
|---|---|---|---|
| LFM | 59.7 | **71.0** | **75.2** |
| InterFlow | 59.7 | 80.0 | 82.4 |

Figure 3: Unconditional image generation by LFM on $32 \times 32$ (i.e., CIFAR10 (upper left) and Imagenet-32 (lower left)) and $128 \times 128$ images (i.e., Flowers (upper right) and LSUN Church (Yu et al., 2015) (lower right)).

Table 4: Success rate (see (27), higher is better) of FM and `LFM` for robotic manipulation on Robomimic (Mandlekar et al., 2021). We evaluate both methods for 100 rollouts and report success rates at different epochs in the format as (Success rate, epochs).

|  | Lift | Can | Square | Transport | Toolhang |
|---|---|---|---|---|---|
| FM | (1.00, 200) | (0.94, 200) | (**0.88**, 200) | (0.60, 200) | (0.52, 200) |
|  |  | (0.98, 500) | (**0.94**, 750) | (0.81, 1500) |  |
| `LFM` | (1.00, 200) | (**0.97**, 200) | (0.87, 200) | (**0.75**, 200) | (**0.53**, 200) |
|  |  | (**0.99**, 500) | (0.93, 750) | (**0.88**, 1500) |  |

(a) Lift    (b) Can    (c) Square    (d) Transport    (e) Toolhang

Figure 4: Robotic manipulation on Robomimic (Mandlekar et al., 2021). **Top row**: initial conditions (IC). **Bottom row**: successful completions. Each task starts from an IC and manipulates the robot arms sequentially to reach successful completion in the end.

## 6.4 ROBOTIC MANIPULATION

We consider robotic manipulation tasks from the Robomimic benchmark (Mandlekar et al., 2021), which consists of 5 tasks of controlling robot arms to perform various pick-and-place operations (Figure 4). E.g., the robot may need to pick up a square object (Figure 4a) or move a soda can from one bin to another (Figure 4b). Recently, generative models that output robotic actions conditioning on the state observations (robot positions or camera image embedding) provide state-of-the-art performance on completing these tasks (Chi et al., 2023). However, it is difficult to transfer pre-trained diffusion or flow models (on natural images) to the conditional generation task here, because the state observations are task-specific and contain nuanced details about the robots, objects, and environment, which are not present in natural images but are necessary to be understood by the model to determine the appropriate actions. Therefore, it is often needed to train a generative model from scratch to directly learn the relation from the task-specific state observations to the actions.

We train a (global) FM model and the proposed `LFM` from scratch, where the total number of parameters is kept the same for both methods. Additional experimental details are in Appendix C.2. As shown in Table 4, `LFM` is competitive against FM in terms of the success rate, and the convergence is faster in some cases (indicated by the higher success rate at early epochs). The performance on the "Toolhang" task does not improve with longer training, as identified in (Chi et al., 2023).

## 7 DISCUSSION

The theoretical analysis can be extended from several angles: First, we analyzed the $\chi^2$ guarantee and induced KL and TV bounds from the former. One may obtain sharper bounds by analyzing KL or TV directly and possibly under weaker assumptions. E.g., Lemma 2.19 in (Albergo et al., 2023) can be used to derive a KL bound for the one-step FM under similar assumptions as our Assumption 2(A2). Second, we currently assume that $T_n^{-1}$ is exact in the reverse process (generation). The analysis can be extended to incorporate inversion error due to numerical computation in practice, possibly by following the strategy in (Cheng et al., 2024). Meanwhile, the proposed methodology has the potential to be further enhanced. In our current experiments, we learn the FM blocks independently parametrized as $\hat{v}(x, t; \theta_n)$. If we introduce weight sharing of $\theta_n$ across $n$, it can impose additional time continuity of the flow model. In addition, it would be useful to explore more distillation techniques under our framework and to extend the approach to more applications.

REPRODUCIBILITY STATEMENT

To ensure reproducibility, we include complete proofs in the Appendix for our theoretical results; For experiments, we describe the dataset, the algorithm detail, and the experimental setup including hyperparameter selections, in the paper and the Appendix. Our reported results of competing baselines are computed using existing code implementation packages that are publicly available online.

ETHICS STATEMENT

There are no potential concerns regarding topics that include, but are not limited to, studies that involve human subjects, practices to data set releases, potentially harmful insights, methodologies and applications, pontential conflicts of interest and sponsorship, discrimination/bias/fairness concerns, privacy and security issues, legal compliance, and research integrity issues.

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

# A PROOFS

Denote by Leb the Lebesgue measure.

**Definition A.1** (Non-degenerate mapping). $T : \mathbb{R}^d \to \mathbb{R}^d$ *is non-degenerate if for any set $A \subset \mathbb{R}^d$ s.t. $Leb(A) = 0$, then $Leb(T^{-1}(A)) = 0$.*

## A.1 PROOFS IN SECTION 4.2

*Proof of Proposition 4.1.* The proof is based on the descending of $\chi^2(p_n\|q)$ in each step. Note that under (A1), by the argument following (13) below, we have $p_0/q \in L^2(q)$ i.e. $\chi^2(p_0\|q) < \infty$.

We first consider the target and learned flows on $[t_{n-1}, t_n]$ to establish the needed descending arguments for the $n$-th step. We shift the time interval to be $[0, T]$, $T := \gamma > 0$, and the transport equations of $\rho$ and $\hat{\rho}$ are as in (9). Define

$$G(t) := \chi^2(\rho_t\|q), \quad \hat{G}(t) := \chi^2(\hat{\rho}_t\|q). \tag{12}$$

By the time endpoint values of $\rho$ and $\hat{\rho}$ as in (9), we have

$$G(0) = \hat{G}(0) = \chi^2(p_{n-1}\|q), \quad G(T) = \chi^2(p_n^*\|q), \quad \hat{G}(T) = \chi^2(p_n\|q).$$

We will show that a) $G(T)$ descends, and b) $\hat{G}(T) \approx G(T)$, then, as a result, $\hat{G}(T)$ also descends.

We first verify the boundedness of $G(t)$ and $\hat{G}(t)$ at all time. By definition, we have (we write integral on $\mathbb{R}^d$ omitting variable $x$ and $dx$ for notation brevity)

$$G(t) = \int \frac{(\rho_t - q)^2}{q} = \int (\frac{\rho_t}{q} - 1)^2 q = \int (\frac{\rho_t}{q})^2 q - 1,$$

when the involved integrals are all finite, and similarly with $\hat{G}(t)$ and $\hat{\rho}_t$. To verify integrability, observe that under (A1),

$$\frac{\rho_t^2}{q}(x), \frac{\hat{\rho}_t^2}{q}(x) \le \frac{C_1^2 e^{-\|x\|^2}}{c_d e^{-\|x\|^2/2}} = \frac{C_1^2}{c_d} e^{-\|x\|^2/2}, \quad c_d := (2\pi)^{-d/2}. \tag{13}$$

This shows that $\rho_t/q, \hat{\rho}_t/q$ and thus $(\rho_t/q - 1), (\hat{\rho}_t/q - 1)$ are all in $L^2(q)$ for all $t$. In particular, this applies to $\rho_0 = \hat{\rho}_0 = p_0$ by taking $n = 1$ and $t = 0$. Thus, $G(t), \hat{G}(t)$ are always finite. Additionally, (13) gives that

$$G(t), \hat{G}(t) \le (C_1/c_d)^2 - 1 \le (C_1/c_d)^2, \quad \forall t \in [0, T], \tag{14}$$

Our descending argument is based on the following two key lemmas:

**Lemma A.2** ($\chi^2$-contraction of OU process). $\chi^2(p_n^*\|q) \le e^{-2\gamma}\chi^2(p_{n-1}\|q)$.

This implies that $G(T) \le e^{-2T}G(0)$.

**Lemma A.3** ($\hat{G}(T) \approx G(T)$). $\chi^2(p_n\|q) \le \chi^2(p_n^*\|q) + C_4\varepsilon^{1/2}$, *with $C_4$ defined in* (20).

Lemma A.2 follows standard contraction results of the diffusion process, and Lemma A.3 utilizes the approximation of $\hat{\rho} \approx \rho$ due to the learning of the velocity field $\hat{v} \approx v$. We postpone the proofs of the two lemmas after the proof of the proposition.

With the two lemmas in hands, we can put the $n$ steps together and prove (11). Define

$$E_n := \chi^2(p_n\|q), \quad \beta := e^{-2\gamma} < 1, \quad \alpha := C_4\varepsilon^{1/2},$$

and then Lemmas A.2-A.3 give that

$$E_n \le \beta E_{n-1} + \alpha.$$

By induction, one can verify that

$$E_n \le \beta^n E_0 + \frac{\alpha(1 - \beta^n)}{1 - \beta} \le \beta^n E_0 + \frac{\alpha}{1 - \beta},$$

which proves (11) and finishes the proof of the proposition. □

*Proof of Lemma A.2.* The lemma follows the well-understood contraction of $\chi^2$ divergence of OU process, see e.g. Bolley et al. (2012). Specifically, on the time interval $[0, T]$, $T = \gamma$, the initial density $p_{n-1}$ renders $\chi^2(p_{n-1}\|q) < \infty$, because $p_{n-1}/q = \rho_0/q \in L^2(q)$ by the argument following (13). For the OU process, since the equilibrium density $q \propto e^{-V}$ with $V(x) = \|x\|^2/2$, we know that $q$ is strongly convex and then satisfies the Poincaré inequality (PI) with constant $C = 1$. By the argument in Eqn. (3) in Bolley et al. (2012), the PI implies the contraction claimed in the lemma. □

*Proof of Lemma A.3.* We prove the lemma by showing the closeness of $\chi^2(p_n\|q) = \hat{G}(T)$ to $\chi^2(p_n^*\|q) = G(T)$. By definition (12),

$$G(t) = \|\frac{\rho_t}{q} - 1\|_{L^2(q)}^2, \quad \hat{G}(t) = \|\frac{\hat{\rho}_t}{q} - 1\|_{L^2(q)}^2, \quad \forall t \in [0, T].$$

We will use the triangle inequality $|\sqrt{G(t)} - \sqrt{\hat{G}(t)}| \le \|\frac{\hat{\rho}_t}{q} - \frac{\rho_t}{q}\|_{L^2(q)}$. Observe that

$$\|\frac{\hat{\rho}_t}{q} - \frac{\rho_t}{q}\|_{L^2(q)}^2 = \int \frac{(\hat{\rho}_t - \rho_t)^2}{q} = \int \frac{(\hat{\rho}_t - \rho_t)^2}{\hat{\rho}_t} \frac{\hat{\rho}_t}{q} \le \frac{C_1}{c_d} \int \frac{(\hat{\rho}_t - \rho_t)^2}{\hat{\rho}_t}, \tag{15}$$

where the inequality is due to the pointwise bound $\frac{\hat{\rho}_t(x)}{q(x)} \le \frac{C_1}{c_d}$, which follows by (A1) and the expression of $q$ with $c_d$ as in (13). We introduce

$$F(t) := \int \frac{(\hat{\rho}_t - \rho_t)^2}{\hat{\rho}_t} = \int (\frac{\rho_t}{\hat{\rho}_t} - 1)^2 \hat{\rho}_t \ge 0, \tag{16}$$

and then we have

$$|\sqrt{G(T)} - \sqrt{\hat{G}(T)}| \le \|\frac{\hat{\rho}_t}{q} - \frac{\rho_t}{q}\|_{L^2(q)} \le (C_1/c_d)^{1/2}\sqrt{F(T)}. \tag{17}$$

Next, we will bound $F(T)$ to be $O(\varepsilon)$, and this will lead to the desired closeness of $\hat{G}(T)$ to $G(T)$.

• Bound of $F(T)$:

We will upper bound $F(T)$ by differentiating $F(t)$ over time. By definition (16), $F(0) = 0$, and (omitting $_t$ in $\rho_t$ and $\hat{\rho}_t$ in the equations)

$$\frac{d}{dt}F(t) = \int (\frac{\rho}{\hat{\rho}} - 1)^2 \partial_t \hat{\rho} + 2(\frac{\rho}{\hat{\rho}} - 1)(\partial_t \rho - \frac{\rho}{\hat{\rho}}\partial_t \hat{\rho})$$

$$= \int 2\partial_t \rho (\frac{\rho}{\hat{\rho}} - 1) - \partial_t \hat{\rho}((\frac{\rho}{\hat{\rho}})^2 - 1)$$

$$= \int -2\nabla \cdot (\rho v)(\frac{\rho}{\hat{\rho}} - 1) + \nabla \cdot (\hat{\rho}\hat{v})((\frac{\rho}{\hat{\rho}})^2 - 1) \quad \text{(by transport equations (9))}$$

$$= \int 2(\rho v) \cdot \nabla(\frac{\rho}{\hat{\rho}}) - (\hat{\rho}\hat{v}) \cdot 2\frac{\rho}{\hat{\rho}}\nabla(\frac{\rho}{\hat{\rho}})$$

$$= 2\int \rho(v - \hat{v}) \cdot \nabla(\frac{\rho}{\hat{\rho}})$$

$$= 2\int (v - \hat{v})\frac{\rho^2}{\hat{\rho}} \cdot (\nabla \log \rho - \nabla \log \hat{\rho}),$$

and the last row is by that $\nabla(\frac{\rho}{\hat{\rho}}) = \frac{\rho}{\hat{\rho}}(\nabla \log \rho - \nabla \log \hat{\rho})$. Thus,

$$\frac{1}{2}F(T) = \frac{1}{2}\int F'(t)dt = \int_0^T \int_{\mathbb{R}^d} (v - \hat{v}) \cdot (\nabla \log \rho - \nabla \log \hat{\rho})\frac{\rho^2}{\hat{\rho}} dx dt.$$

Then, by Cauchy-Schwarz, we have

$$\frac{1}{2}F(T) \le \left(\int_0^T \int_{\mathbb{R}^d} \|v - \hat{v}\|^2 \rho dx dt\right)^{1/2} \left(\int_0^T \int_{\mathbb{R}^d} \|\nabla \log \rho - \nabla \log \hat{\rho}\|^2 \frac{\rho^3}{\hat{\rho}^2} dx dt\right)^{1/2}. \tag{18}$$

By the flow-matching error assumption (A0), we have the first factor in the r.h.s. of (18) bounded by $\varepsilon$. Meanwhile, the technical conditions (A2)(A3) together imply that for all $t \in [0, T]$,

$$\int_{\mathbb{R}^d} \|\nabla \log \rho_t - \nabla \log \hat{\rho}_t\|^2 (\rho_t^3 / \hat{\rho}_t^2) dx \le (2L)^2 \int_{\mathbb{R}^d} (1 + \|x\|)^2 (\rho_t^3 / \hat{\rho}_t^2)(x) dx \le (2L)^2 C_2,$$

thus the second factor in (18) is upper bounded by $\sqrt{(2L)^2 C_2 T}$. Putting together, we have

$$F(T) \le 2(2L)\sqrt{C_2 T}\varepsilon = C_3 \varepsilon, \quad C_3 := 4L\sqrt{C_2 \gamma}. \tag{19}$$

• Bound of $\hat{G}(T) - G(T)$:

With the bound (19) of $F(T)$, we are ready to go back to (17), which gives

$$\sqrt{\hat{G}(T)} \le \sqrt{G(T)} + (C_1/c_d)^{1/2}\sqrt{F(T)}.$$

Together with that $G(T) \le (C_1/c_d)^2$ by (14), this gives that

$$\hat{G}(T) \le G(T) + 2(C_1/c_d)^{3/2}\sqrt{C_3}\varepsilon^{1/2} + (C_1/c_d)C_3\varepsilon \le G(T) + C_4\varepsilon^{1/2},$$

where we used that $\varepsilon < 1$ by (A0) and

$$C_4 := 2(C_1/c_d)^{3/2}\sqrt{C_3} + (C_1/c_d)C_3. \tag{20}$$

The fact that $\hat{G}(T) \le G(T) + C_4\varepsilon^{1/2}$ proves the lemma. $\qquad\square$

## A.2 PROOFS IN SECTION 4.3

**Lemma A.4** (Bi-direction DPI). *Let* $D_f$ *be an* $f$-*divergence. If* $T : \mathbb{R}^d \to \mathbb{R}^d$ *is invertible and for two densities* $p$ *and* $q$ *on* $\mathbb{R}^d$, $T_\# p$ *and* $T_\# q$ *also have densities, then*

$$D_f(p\|q) = D_f(T_\# p\|T_\# q).$$

*Proof of Lemma A.4.* Let $X_1 \sim p$, $X_2 \sim q$, and $Y_1 = T(X_1)$, $Y_2 = T(X_2)$. Then $Y_1$ and $Y_2$ also have densities, $Y_1 \sim \tilde{p} := T_\# p$ and $Y_2 \sim \tilde{q} := T_\# q$. By the classical DPI of $f$-divergence (see, e.g., the introduction of Raginsky (2016)), we have $D_f(\tilde{p}\|\tilde{q}) \le D_f(p\|q)$. In the other direction, $X_i = T^{-1}(Y_i)$, $i = 1, 2$, then DPI also implies $D_f(p\|q) \le D_f(\tilde{p}\|\tilde{q})$. $\qquad\square$

*Proof of Theorem 4.2.* Under the assumptions, Proposition 4.1 applies to give that

$$\chi^2(p_N\|q) \le e^{-2\gamma N}\chi^2(p_0\|q) + \frac{C_4}{1 - e^{-2\gamma}}\varepsilon^{1/2}.$$

Then, whenever $e^{-2\gamma N}\chi^2(p_0\|q) \le \varepsilon^{1/2}$ which is ensured by the $N$ in the theorem, we have $\chi^2(p_N\|q) \le (1 + \frac{C_4}{1 - e^{-2\gamma}})\varepsilon^{1/2}$. Let $T_1^N := T_N \circ \cdots \circ T_1$, which is invertible, and $p_N = (T_1^N)_\# p_0$, $q_N = (T_1^N)_\# q_0$. We will apply the bi-directional DPI Lemma A.4 to show that

$$\chi^2(p_0\|q_0) = \chi^2((T_1^N)_\# p_0\|(T_1^N)_\# q_0) = \chi^2(p_N\|q_N) = \chi^2(p_N\|q), \tag{21}$$

which then proves the theorem.

For Lemma A.4 to apply to show the first equality in (21), it suffices to verify that $p_0, q_0, p_N, q_N$ all have densities. $p_0$ has density by the theorem assumption. From Definition A.1, one can verify that a transform $T$ being non-degenerate guarantees that $P$ has density $\Rightarrow T_\# P$ has density (see, e.g., Lemma 3.2 of (Cheng et al., 2024)). Because all $T_n$ are non-degenerate under (A0), from that $p_0$ has density we know that all $p_n$ has densities, including $p_N$. In the reverse direction, $q_N = q$ which is the normal density. By that $T_n^{-1}$ are all non-degenerate, we similarly have that all $q_n$ has densities, including $q_0$. $\qquad\square$

**Lemma A.5.** *For two densities* $p$ *and* $q$ *where* $\chi^2(p\|q) < \infty$, $\mathrm{KL}(p\|q) \le \chi^2(p\|q)$.

*Proof.* The statement is a well-known fact and we include an elementary proof for completeness. Because $\chi^2(p\|q) < \infty$, we have $(p/q - 1)$ and thus $p/q \in L^2(q)$, i.e. $\int p^2/q < \infty$. By the fact that $\log x \leq x - 1$ for any $x > 0$, $\log \frac{p}{q}(x) \leq \frac{p}{q}(x) - 1$, and then

$$\mathrm{KL}(p\|q) = \int p \log \frac{p}{q} \leq \int p(\frac{p}{q} - 1) = \int \frac{p^2}{q} - 1 = \chi^2(p\|q).$$

$\square$

*Proof of Corollary 4.3.* Let $\rho_t = (\mathrm{OU})_0^t(P)$. Because $M_2(P) < \infty$, one can show that $\mathcal{W}_2(\rho_t, P)^2 = O(t)$ as $t \to 0$. Specifically, by Lemma C.1 in (Cheng et al., 2024), $\mathcal{W}_2(\rho_t, P)^2 \leq t^2 M_2(P) + 2td$. Thus, for $t < 1$, $\mathcal{W}_2(\rho_t, P)^2 \leq (M_2(P) + 2d)t$. This proves $\mathcal{W}_2(P, \rho_\delta) \leq C_5 \delta^{1/2}$.

Meanwhile, $p_0 = \rho_\delta$ satisfies the needed condition in Theorem 4.2, the claimed bound of $\chi^2(\rho_\delta\|q_0)$ directly follows from Theorem 4.2.

$\square$

**Lemma A.6.** *Suppose $P$ on $\mathbb{R}^d$ is compactly supported, then, $\forall t > 0$, $\rho_t = (\mathrm{OU})_0^t(P)$ satisfies satisfies (10) for some $C_1$, $C_2$ and $L$ (which may depend on $t$).*

*Proof of Lemma A.6.* Suppose $P$ is supported on $B_R := \{x \in \mathbb{R}^d, \|x\| \leq R\}$ for some $R > 0$.

By the property of the OU process, $\rho_t$ is the probability density of the random vector

$$Z_t := e^{-t} X_0 + \sigma_t Z, \quad Z \sim \mathcal{N}(0, I_d), \quad X_0 \sim P, \quad Z \perp X_0. \tag{22}$$

where $\sigma_t^2 = 1 - e^{-2t}$. We will verity the first two inequalities in (10), and the third one is implied by the first one. (The third one also has a direct proof: $M_2(\rho_\delta) = \mathbb{E}\|Z_\delta\|^2 = e^{-2\delta} M_2(P) + \sigma_\delta^2 d < \infty$, then the third condition in (10) holds with $C_2 = 1 + e^{-2\delta} M_2(P) + \sigma_\delta^2 d$.)

The law of $Z_t$ in (22) gives that

$$\rho_t(x) = \int_{\mathbb{R}^d} \frac{1}{(2\pi\sigma_t^2)^{d/2}} e^{-\|x - e^{-t}y\|^2/(2\sigma_t^2)} dP(y). \tag{23}$$

This allows us to verity the first two inequalities in (10) with proper $C_1$ and $L$, making use of the fact that $P$ is supported on $B_R$. Specifically, by definition,

$$\nabla \rho_t(x) = \int_{\mathbb{R}^d} \frac{1}{(2\pi\sigma_t^2)^{d/2}} \left( -\frac{x - e^{-t}y}{\sigma_t^2} \right) e^{-\|x - e^{-t}y\|^2/(2\sigma_t^2)} dP(y),$$

and then, because $\|x - e^{-t}y\| \leq \|x\| + e^{-t}\|y\| \leq \|x\| + e^{-t}R$,

$$\|\nabla \rho_t(x)\| \leq \frac{\|x\| + e^{-t}R}{\sigma_t^2} \int_{\mathbb{R}^d} \frac{1}{(2\pi\sigma_t^2)^{d/2}} e^{-\|x - e^{-t}y\|^2/(2\sigma_t^2)} dP(y) = \frac{\|x\| + e^{-t}R}{\sigma_t^2} \rho_t(x).$$

This means that

$$\frac{\|\nabla \rho_t(x)\|}{\rho_t(x)} \leq \frac{\|x\| + e^{-t}R}{\sigma_t^2},$$

which means that the 2nd condition in (10) holds with $L = \frac{1}{\sigma_t^2} \max\{1, e^{-t}R\}$.

To prove the first inequality, we again use the expression (23). By that $\|x\| \leq \|x - e^{-t}y\| + \|e^{-t}y\|$, and that $\|y\| \leq R$, we have

$$\|x\|^2 \leq \|x - e^{-t}y\|^2 + 2\|x - e^{-t}y\|\|e^{-t}y\| + \|e^{-t}y\|^2$$
$$\leq \|x - e^{-t}y\|^2 + 2(\|x\| + e^{-t}R)e^{-t}R + e^{-2t}R^2$$
$$= \|x - e^{-t}y\|^2 + 2e^{-t}R\|x\| + 3e^{-2t}R^2,$$

and thus

$$e^{-\|x - e^{-t}y\|^2/(2\sigma_t^2)} \leq e^{-\frac{1}{2\sigma_t^2}(\|x\|^2 - 2e^{-t}R\|x\| - 3e^{-2t}R^2)}$$
$$= e^{-\frac{1}{2\sigma_t^2}\|x\|^2 + \frac{e^{-t}R}{\sigma_t^2}\|x\|} e^{\frac{3e^{-2t}R^2}{2\sigma_t^2}}.$$

Inserting in (23), we have that

$$\rho_t(x) \leq \alpha_t e^{-\frac{1}{2\sigma_t^2}\|x\|^2 + \frac{e^{-t}R}{\sigma_t^2}\|x\|}, \quad \alpha_t := \frac{e^{\frac{3e^{-2t}R^2}{2\sigma_t^2}}}{(2\pi\sigma_t^2)^{d/2}}.$$

For $\rho_t(x) \leq C_1 e^{-\|x\|^2/2}$, it suffices to have $C_1$ s.t.

$$\frac{C_1}{\alpha_t} \geq e^{\frac{1}{2}(1-\frac{1}{\sigma_t^2})\|x\|^2 + \frac{e^{-t}R}{\sigma_t^2}\|x\|}. \tag{24}$$

Because $t > 0$, $1 - \frac{1}{\sigma_t^2} = \frac{-e^{-2t}}{1-e^{-2t}} < 0$, the r.h.s. of (24) as a function on $\mathbb{R}^d$ decays faster than $e^{-c\|x\|^2}$ for some $c > 0$ as $\|x\| \to \infty$, and then the function is bounded on $\mathbb{R}^d$. This means that there is $C_1 > 0$ to make (24) hold. This proves that the first inequality in (10) holds for $\rho_t$. $\qquad\square$

# B  DETAILS OF LFM SCHEDULE AND NLL COMPUTATION

## B.1  SELECTION OF LFM SCHEDULE

Given a positive integer $N \geq 1$, we specify $\{\gamma_n\}_{n=1}^{N-1}$ via the following scheme:

$$\gamma_n = \rho^{n-1}c, n = 1, 2, \ldots \tag{25}$$

where the base time stamp $c$ and the multiplying factor $\rho$ are user-specified hyper-parameters.

## B.2  NEGATIVE LOG-LIKELIHOOD COMPUTATION

Based on the instantaneous change-of-variable formula in neural ODE Chen et al. (2018), we know that for a trained flow model $\hat{v}(x(t), t; \theta)$ on $[0, 1]$ that interpolates between $p$ and $q$, the log-likelihood of data $x \sim p$ can be expressed as

$$\log p(x) = \log q(x(1)) + \int_0^1 \nabla \cdot \hat{v}(x(s), s; \theta)ds, x(0) = x,$$

where $\nabla \cdot \hat{v}(\cdot, s; \theta)$ is the trace of the Jacobian matrix of the network function.

Our trained LFM has $N$ sub-flows $\hat{v}_n(\cdot, t; \theta)$ for $n = 1, \ldots, N$. The log-likelihood at test sample $x \sim p$ is then computed by

$$\log p(x) = \log q(x_N) + \sum_{n=1}^{N} \int_0^1 \nabla \cdot \hat{v}_n(x(s), s; \theta)ds, \tag{26}$$

where starting from $x_0 = x$,

$$x_n = x_{n-1} + \int_0^1 \hat{v}_n(x(s), s; \theta)ds, \quad x(0) = x_{n-1}$$

Both the integration of $\hat{v}_n$ and $\nabla \cdot \hat{v}_n$ are computed using the numerical scheme of neural ODE. We report NLL in $\log_e$ (known as "nats") in all our experiments.

---

**Algorithm A.1** Distillation of LFM

---

**Input:** Samples $\sim q_n$, $n = 0, \cdots, N$, generated by a pre-trained $N$-block LFM

**Output:** $N' = N/k$ distilled sub-models $\{T_n^D(\cdot)\}_{n=1}^{N'}$.

1: **for** $n = 1, \ldots, N'$ **do**
2:     Train $f(x; \theta_n^D)$ via $\min_{\theta_n^D} \mathbb{E}_{(x_n, x_{n-1}) \sim (q_{N-kn}, q_{N-k(n-1)})} \|(x_n - x_{n-1}) - f(x_{n-1}; \theta_n^D)\|^2$.
3:     Output $T_n^D(x) = x + f(x; \theta_n^D)$.
4: **end for**

---

## C  EXPERIMENTAL DETAILS AND ADDITIONAL RESULTS

To train LFM sub-flows, we use the trigonometric (Trig) interpolant on 2D, tabular, and image generation experiments (Sections 6.1–6.3) and use the optimal transport interpolant on robotic manipulation (Section 6.4). During inference per block, we employ the Dormand-Prince-Shampine ODE sampler with tolerances of 1e-5 for 2d and tabular data, 1e-4 for 32x32 images, and 1e-3 for 128x128 images. We use 1 Euler step on the robotic manipulation experiments as prior works have shown the inference efficiency of FM on such tasks (Hu et al., 2024).

### C.1  2D, TABULAR, AND IMAGE EXPERIMENTS

In all experiments, we use the Adam optimizer (Kingma, 2014) with the following parameters: $\beta_1 = 0.9, \beta_2 = 0.999, \epsilon = 1e-8$; the learning rate is to be specified in each experiment. Additionally, the number of training batches indicates how many batches pass through all $N$ sub-flows per Adam update.

On two-dimensional datasets and tabular datasets, we parameterize local sub-flows with fully-connected networks; the dataset detail and hyper-parameters of LFM are in Table A.1.

On image generation examples, we parameterize local sub-flows as UNets (Nichol & Dhariwal, 2021), where the dataset details and training specifics are provided in Table A.2.

### C.2  ROBOTIC MANIPULATION

In the context of generative modeling, this task of robotic manipulation can be understood as performing sequential conditional generation. Specifically, at each time step $t \geq 1$, the goal is to model the conditional distribution $A_t|O_t$, where $O_t \in \mathbb{R}^O$ denotes the states of the robots at time $t$ and $A_t \in \mathbb{R}^A$ is the action that controls the robots. During inference, the robot is controlled as we iteratively sample from $A_t|O_t$ across time steps $t$. Past works leveraging diffusion models have reached state-of-the-art performances on this task (Chi et al., 2023), where a neural network $v_\theta$ (e.g., CNN-based UNet (Janner et al., 2022)) is trained to approximate the distribution $A_t|O_t$ via DDPM (Ho et al., 2020). More recently, flow-based methods have also demonstrated competitive performance with faster inference (Hu et al., 2024).

We use the widely adopted *success rate* to examine the performance of a robot manipulator:

$$\text{Success rate} = \frac{\#\text{success rollouts}}{\#\text{rollouts}} \in [0, 1]. \tag{27}$$

Specifically, starting from a given initial condition $O_1$ of the robot, each rollout denotes a trajectory $S = \{O_1, A_1, O_2, \ldots, A_T, O_T\}$ where $A_t|O_t$ is modeled by the generative. The rollout $S$ is a success if at any $t \in 1, \ldots, T$, the robotic state $O_t$ meets the success criterion (e.g., successfully pick up the square as in the task "lift" in Figure 4a).

We also describe details of each of the 5 Robomimic tasks below, including dimensions of observations $O_t$ and actions $A_t$ and the success criteria. The initial condition and final successful completion were shown in Figure 4. Table A.3 contains the hyper-parameter setting in each task, where we use the same network and training procedure as in (Chi et al., 2023).

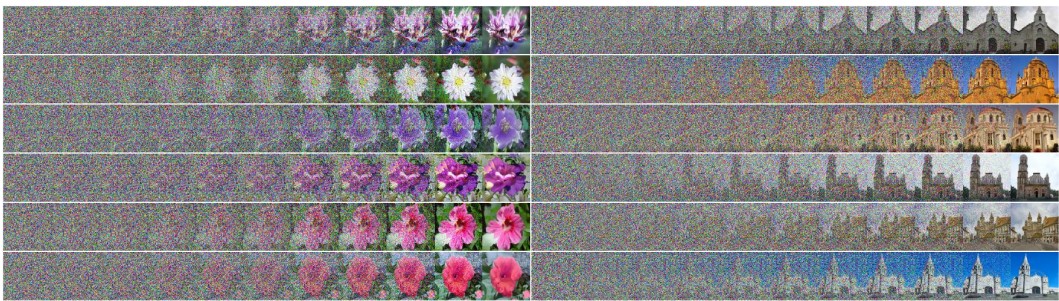

Figure A.1: Noise-to-image trajectories by LFM: Flowers (left) and LSUN Church (right).

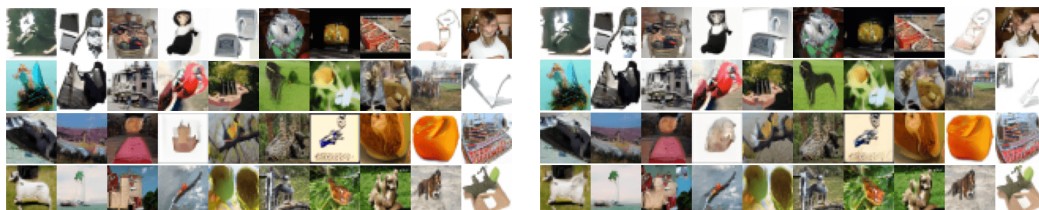

| (a) `LFM` before distillation. | (b) `LFM` after distillation. |

Figure A.2: Unconditional image generation on Imagenet-32 before and after distillation. We distill `LFM` into a 5-NFE model.

|  | Rose | Fractal tree | POWER | GAS | MINIBOONE | BSDS300 |
|---|---|---|---|---|---|---|
| Dimension | 2 | 2 | 6 | 8 | 43 | 63 |
| # Training point | 2,000,000 | 2,000,000 | 1,615,917 | 852,174 | 29,556 | 1,000,000 |
| Batch Size | 10K | 10K | 30K | 50K | 1000 | 500 |
| Training Batches | 50K | 50K | 100K | 100K | 100K | 30K |
| Hidden layer width (per sub-flow) | 256 | 256 | 256 | 362 | 362 | 512 |
| # Hidden layers | 3 | 3 | 4 | 5 | 4 | 4 |
| Activation | Softplus | Softplus | ReLU | ReLU | ReLU | ELU |
| # sub-flows $N$ | 9 | 9 | 4 | 2 | 2 | 4 |
| $(c, \rho)$ in (25) | (0.025, 1.25) | (0.025, 1.25) | (0.15, 1.3) | (0.05, 1) | (0.35, 1) | (0.25, 1) |
| Total # parameters in M (all sub-flows) | 1.20 | 1.20 | 0.81 | 1.06 | 0.85 | 3.41 |
| Learning Rate (LR) | 0.0002 | 0.0002 | 0.005 | 0.002 | 0.005 | 0.002 |
| LR decay (factor, frequency in batches) | (0.99, 1000) | (0.99, 1000) | (0.99, 1000) | (0.99, 1000) | (0.9, 4000) | (0.8, 4000) |
| Beta $\alpha, \beta$, time samples | (1.0, 1.0) | (1.0, 1.0) | (1.0, 1.0) | (1.0, 0.5) | (1.0, 1.0) | (1.0, 1.0) |

Table A.1: Hyperparameters and architecture for two-dimensional datasets and tabular datasets. The table is formatted similarly as (Albergo & Vanden-Eijnden, 2023, Table 3).

**Lift:** The goal is for the robot arm to lift a small cube in red. Each $O_t$ has dimension $16 \times 19$ and each $A_t$ has dimension $16 \times 10$, representing state-action information for the next 16 time steps starting at $t$.

**Can:** The goal is for the robot to pick up a coke can from a large bin and place it into a smaller target bin. Each $O_t$ has dimension $16 \times 23$ and each $A_t$ has dimension $16 \times 10$, representing state-action information for the next 16 time steps starting at $t$.

**Square:** The goal is for the robot to pick up a square nut and place it onto a rod with precision. Each $O_t$ has dimension $16 \times 23$ and each $A_t$ has dimension $16 \times 10$, representing state-action information for the next 16 time steps starting at $t$.

**Transport:** The goal is for the two robot arms to transfer a hammer from a closed container on one shelf to a target bin in another shelf. Before placing the hammer, one arm has to also clear the target bin by moving away a piece of trash to the nearby receptacle. The hammer must be picked up by one

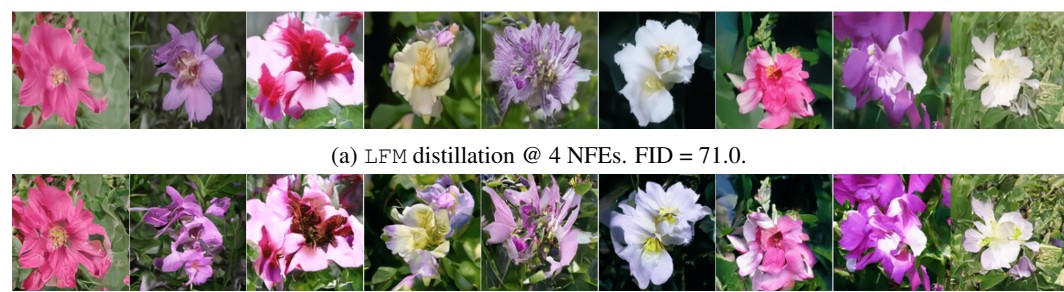

(a) `LFM` distillation @ 4 NFEs. FID = 71.0.

(b) InterFlow distillation @ 4 NFEs. FID = 80.0.

Figure A.3: Qualitative comparison of `LFM` and InterFlow after distillation.

|  | CIFAR-10 | Imagenet-32 | Flowers | LSUN Churches |
|---|---|---|---|---|
| Dimension | 32×32 | 32×32 | 128×128 | 128×128 |
| # Training point | 50,000 | 1,281,167 | 8,189 | 126,227 |
| Batch Size | 200 | 256 | 40 | 40 |
| Training Batches | $5 \times 10^4$ | $2 \times 10^5$ | $4 \times 10^4$ | $1.2 \times 10^5$ |
| Hidden dim (per sub-flow) | 128 | 114 | 128 | 128 |
| # sub-flows $N$ | 4 | 5 | 4 | 4 |
| $(c, \rho)$ in (25) | (0.3, 1.1) | (0.3, 1.1) | (0.5, 1.5) | (0.4, 1.5) |
| Total # parameters in M (all sub-flows) | 160 | 120 | 464 | 464 |
| Learning Rate (LR) | 0.0001 | 0.0001 | 0.0002 | 0.0002 |
| U-Net dim mult | [1,2,2,2,2] | [1,2,2,2] | [1,1,2,3,4] | [1,1,2,3,4] |
| Beta $\alpha, \beta$, time samples | (1.0, 1.0) | (1.0, 1.0) | (1.0, 1.0) | (1.0, 1.0) |
| Learned $t$ sinusoidal embedding | Yes | Yes | Yes | Yes |
| # GPUs | 1 | 1 | 1 | 1 |

Table A.2: Hyperparameters and architecture for image datasets. The table is formatted similarly as (Albergo & Vanden-Eijnden, 2023, Table 4).

|  | Lift | Can | Square | Transport | Toolhang |
|---|---|---|---|---|---|
| Batch Size | 256 | 256 | 256 | 256 | 256 |
| Training Epochs | 200 | 500 | 750 | 1500 | 200 |
| Hidden dims (per sub-flow) | [128,256,512] | [128,256,512] | [128,256,512] | [176,352,704] | [128,256,512] |
| # sub-flows $N$ | 4 | 4 | 4 | 2 | 4 |
| $(c, \rho)$ in (25) | (0.15, 1.25) | (0.2, 1.25) | (0.2, 1) | (0.5, 1) | (0.25, 1.25) |
| Total # parameters in M (all sub-flows) | 66 | 66 | 66 | 67 | 67 |
| Learning Rate (LR) | 0.0001 | 0.0001 | 0.0001 | 0.0001 | 0.0001 |
| # GPUs | 1 | 1 | 1 | 1 | 1 |

Table A.3: Hyperparameters and architecture for robotic manipulation under state-based environment on Robomimic (Mandlekar et al., 2021).

arm which then hands over to the other. Each $O_t$ has dimension $16 \times 59$ and each $A_t$ has dimension $16 \times 20$, representing state-action information for the next 16 time steps starting at $t$.

**Toolhang:** The goal is for the robot to assemble a frame that includes a base piece and a hook piece by inserting the hook into the base. The robot must then hang a wrench on the hook. Each $O_t$ has dimension $16 \times 53$ and each $A_t$ has dimension $16 \times 10$, representing state-action information for the next 16 time steps starting at $t$.

