# OpenReview forum: "Local Flow Matching Generative Models"
_ICLR.cc/2025/Conference — Submitted to ICLR 2025_

### Official Review · Reviewer_Snd7 · 2024-10-21

**Soundness:** 4
**Presentation:** 3
**Contribution:** 3
**Rating:** 6
**Confidence:** 3

**Summary:**

The paper proposes a novel generative model, Local Flow Matching (LFM), which segments Flow Matching along the time dimension. The transition between two distributions within each time segment is modeled using a smaller-scale sub-model. This generative model offers the advantages of having smaller sub-models, faster convergence during training, and greater convenience for distillation.

**Strengths:**

The paper provides a detailed theoretical proof that the data distribution generated by LFM is close to the true data distribution, demonstrating the exponential convergence of the forward process and the generation guarantee of the backward process. In addition, the paper conducts extensive experiments to validate the effectiveness of LFM, including generation on two-dimensional toy data, tabular data generation, image generation, and robotic manipulation. Moreover, the paper is well-written.

**Weaknesses:**

Some experimental details in the paper may need further clarification. The data in Table 2 is used to demonstrate that LFM requires less computational cost, but the model sizes of both LFM and InterFlow are not provided. Additionally, it is unclear whether the training data shown in the table refers to the data used by each sub-model individually or by all sub-models collectively.

**Questions:**

Could you provide more detailed data regarding the issues mentioned in the weaknesses, such as the specific number of parameters for each sub-model in LFM, the number of sub-models, and the parameter count of the InterFlow model (or explain why the total number of parameters across all sub-models is equal to that of InterFlow). Could you provide the amount of data required to train each sub-model?

---

> ### Author Response · Authors · 2024-11-25
> **Response to Reviewer Snd7**
>
> 1. **(Weakness) Some experimental details in the paper may need further clarification. The data in Table 2 is used to demonstrate that LFM requires less computational cost, but the model sizes of both LFM and InterFlow are not provided. Additionally, it is unclear whether the training data shown in the table refers to the data used by each sub-model individually or by all sub-models collectively.**
>
> Please refer to the second question in the common response regarding the same number of model parameters being used throughout experiments. Meanwhile, the batch size in Table 2 refers to the number of data used by each sub-model individually, each of which is $K$ times smaller than the global flow, assuming $K$ local flows are used in LFM.
>
> 2. **(Question) Could you provide more detailed data regarding the issues mentioned in the weaknesses, such as the specific number of parameters for each sub-model in LFM, the number of sub-models, and the parameter count of the InterFlow model (or explain why the total number of parameters across all sub-models is equal to that of InterFlow). Could you provide the amount of data required to train each sub-model?**
>
> Information regarding “the specific number of parameters for each sub-model in LFM, the number of sub-models, and the parameter count of the InterFlow model” is provided in Appendix Tables A.1 - A.3. The total number of parameters across all sub-models is equal to that of InterFlow because we scale the width of hidden layers of LFM, so that each block is proportionally smaller.
>
>
> The amount of data required to train each sub-model is the same as that used to train the global flow. More specifically, training data of the $n$-block is the empirical push-forward distribution by the $(n-1)$-th block, which maps its training data through its ODE solution map to obtain the push-forward distribution.

---

### Official Review · Reviewer_vDPz · 2024-10-26

**Soundness:** 2
**Presentation:** 3
**Contribution:** 2
**Rating:** 3
**Confidence:** 4

**Summary:**

The paper introduces Local Flow Matching (LFM), a generative approach that builds upon the existing Flow Matching (FM) framework. LFM improves upon FM by dividing the global flow into multiple local flow sub-models. Each sub-model is trained to match a diffusion process between closer intermediate distributions, which reduces the model size and accelerates training.

**Strengths:**

1.	The paper offers a fresh approach in the field of generative modeling, combining ideas from diffusion processes and FM. The idea of breaking down a single large flow into several smaller steps (local flows) is natural.
2.	The paper provides solid theoretical guarantees, specifically proving a generation guarantee in terms of the $\chi^2$-divergence between the generated and true data distributions. The experiments are comprehensive, covering a range of tasks from image generation to robotic manipulation, and the results demonstrate the performance of LFM.

**Weaknesses:**

1.	FM simplifies the diffusion process into a single step, transforming a trajectory from a curve into a straight line, thus reducing training costs and improving sampling efficiency. However, this paper reverses that advantage by breaking this single step into multiple segments. The division of the trajectory into multiple steps could introduce added computational complexity, potentially negating the efficiency gains that FM originally aimed to provide.

2.	The paper presents some conceptual ambiguities regarding key terms. Continuous Normalizing Flows (CNF) and neural ODEs are two distinct models, but the paper incorrectly states that "CNF trains a neural ODE" (Line 106). In fact, FM is a method for training CNF models, not a model in itself, much like the proposed LFM method. The paper conflates these concepts in several places (e.g. Line 41, 51), which could cause confusion.

3.	The paper highlights improved training efficiency through the use of smaller local models. However, the exact speed improvements are not always well-quantified. For example, while the paper states that LFM leads to faster convergence and reduced memory usage, these claims could be supported by more detailed analysis. Providing concrete timing metrics (e.g., runtime comparisons, memory benchmarks across different hardware) would bolster the argument for LFM's practical benefits. More information can be found in Questions part.

4.	In line 127, a reference is used as the subject of the sentence but is enclosed in parentheses. Please correct this formatting issue and ensure consistency throughout the paper.

**Questions:**

1.	In the training process, at each step, the target distribution is generated from the sample of the previous step through the Ornstein-Uhlenbeck (OU) process. What is the theoretical basis for using the OU process in this context? This essentially defines checkpoints along the probability path. Would a different process yield similar results, or is there a specific reason the OU process was chosen?

2.	In Appendix B1, how were the stepwise parameters $\rho$ and $c$ selected? Were they chosen solely based on extensive experimentation and selecting the ones that gave the best results, or was there a more principled approach to their selection?

3.	In the image generation tasks (Table 2), it would be beneficial to include comparisons with Flow Matching (FM) [1], rather than only with InterFlow, as Local Flow Matching (LFM) is based on FM. This would better illustrate the claimed improved training efficiency. Additionally, either in the main text or in the appendix, it would be helpful to provide a comparison of the total number of parameters for each method across all experiments, rather than only ensuring the same training scheme.

[1] Yaron Lipman, Ricky T. Q. Chen, Heli Ben-Hamu, Maximilian Nickel, and Matthew Le. Flow matching for generative modeling. In The Eleventh International Conference on Learning Representations, 2023.

---

> ### Author Response · Authors · 2024-11-25
> **Response to Reviewer vDPz**
>
> Please see our first question in the common response regarding the comparison of LFM with FM.
>
> 1. **(Weakness) FM simplifies the diffusion process into a single step, transforming a trajectory from a curve into a straight line, thus reducing training costs and improving sampling efficiency. However, this paper reverses that advantage by breaking this single step into multiple segments. The division of the trajectory into multiple steps could introduce added computational complexity, potentially negating the efficiency gains that FM originally aimed to provide.**
>
> We appreciate the comment and address it below:
> - "FM...transforming a trajectory from a curve into a straight line...thus reducing training costs and improving sampling efficiency": Theoretically, as argued in (Lipman et al., 2023), the linear interpolation by FM is the optimal transport map, leading to "straight line trajectories with constant speed". However, we empirically found that the trained velocity field yields curvy trajectories when the data distribution $P$ is close to the target Gaussian distribution. We tested this by setting $P=\mathcal{N}(\mu, I_2)$ for $\mu=[-0.05, -0.05]$ and training an FM model $v_{\theta}$. Visualizing the ODE trajectory (see [here](https://anonymous.4open.science/r/LFM_ICLR-B14C/trajectory_FM.png)) given 20 random starting points using $v_{\theta}$ shows that the trajectory can be highly curvy. Thus, it remains unclear under what conditions the empirically trained FM model yields straight trajectories and consequently "reduces training costs and improves sampling efficiency" as suggested.
> - "The division of the trajectory into multiple steps could introduce added computational complexity, potentially negating the efficiency gains that FM originally aimed to provide": Having addressed why FM does not necessarily yield straight trajectories and thus establish efficiency gains, we argue that dividing the trajectory into multiple steps does not necessarily introduce added computational complexity. Our experiments consistently show improved generation performance under the same number of training batches and total number of model parameters using LFM. Additionally, using the 2D rose dataset, we demonstrate that the ODE trajectory for each LFM block (under OT interpolant) is fairly straight (see [here](https://anonymous.4open.science/r/LFM_ICLR-B14C/trajectory_LFM.png)). This is achieved by training each LFM block using a *dependent* coupling of $(p_{n-1}, p_n^*)$. Specifically, given the left end $x_l'\sim p_{n-1}$ and an independent $g \sim \mathcal{N}(0,I_d)$, the right end $x_r$ as defined in Eq. (6) depends on $x_l'$.
>
> 2. **(Weakness) The paper presents some conceptual ambiguities regarding key terms. Continuous Normalizing Flows (CNF) and neural ODEs are two distinct models, but the paper incorrectly states that "CNF trains a neural ODE" (Line 106). In fact, FM is a method for training CNF models, not a model in itself, much like the proposed LFM method. The paper conflates these concepts in several places (e.g. Line 41, 51), which could cause confusion.**
>
> We appreciate the reviewer's comments and agree with their observations. We have revised our statements to address these points:
> - Regarding "CNF trains a neural ODE": We acknowledge the ambiguity here, which may imply CNF is a method to train a neural ODE. To clarify, CNF uses a neural ODE as its underlying model, which is optimized by maximizing the data likelihood. We have revised this statement in the paper to reflect this more accurate description.
> - Regarding FM and LFM: We agree that FM was originally introduced as "a simulation-free approach for training CNFs" rather than a model itself. In this context, our proposed LFM is best understood as an improved version of FM for training CNF models. This clarification helps to position both FM and LFM as methods for training CNFs, rather than standalone models.
>
> 3. **(Weakness) In line 127, a reference is used as the subject of the sentence but is enclosed in parentheses. Please correct this formatting issue and ensure consistency throughout the paper.**
>
> We have fixed this issue and similar ones in the paper.
>
> 4. **(Question) In the training process, at each step, the target distribution is generated from the sample of the previous step through the Ornstein-Uhlenbeck (OU) process. What is the theoretical basis for using the OU process in this context? This essentially defines checkpoints along the probability path. Would a different process yield similar results, or is there a specific reason the OU process was chosen?**
>
> Please refer to Section 4 for in-depth theoretical analyses on OU process benefits. The OU process ensures:
> - Exponential convergence of the forward process in $\chi^2$ divergence (Proposition 4.1)
> - Generation guarantee for distributions with regular data density (Theorem 4.2)
>
> While alternative processes may be possible, their analysis is beyond this work's scope.

---

> > ### Author Response · Authors · 2024-11-25
> > **(Cont) Response to Reviewer vDPz**
> >
> > 5. **(Question) In Appendix B1, how were the stepwise parameters selected? Were they chosen solely based on extensive experimentation and selecting the ones that gave the best results, or was there a more principled approach to their selection?**
> >
> > We designed the stepwise parameter based on an exponential scheme $\gamma_n = \rho^{n-1}c$, which is chosen because of the exponential convergence of the forward process (see Proposition 4.1). In reality, the best choice of $(\rho, c)$ varies across experiments, yet we often found $\rho \in [0.1, 0.25]$ and $c \in [1.5, 2]$ to work reasonably well for all settings in LFM.
> >
> > 6. **(Question) In the image generation tasks (Table 2), it would be beneficial to include comparisons with Flow Matching (FM) [1], rather than only with InterFlow, as Local Flow Matching (LFM) is based on FM. This would better illustrate the claimed improved training efficiency. Additionally, either in the main text or in the appendix, it would be helpful to provide a comparison of the total number of parameters for each method across all experiments, rather than only ensuring the same training scheme.
> > [1] Yaron Lipman, Ricky T. Q. Chen, Heli Ben-Hamu, Maximilian Nickel, and Matthew Le. Flow matching for generative modeling. In The Eleventh International Conference on Learning Representations, 2023.**
> >
> > Please refer to the first question in the common response for LFM vs. FM comparison. We demonstrated enhanced training efficiency across both image and non-image examples.
> >
> > Please refer to the second question in the common response regarding the same number of model parameters being used throughout experiments.
> >
> > 7. **(Weakness) The paper highlights improved training efficiency through the use of smaller local models. However, the exact speed improvements are not always well-quantified. For example, while the paper states that LFM leads to faster convergence and reduced memory usage, these claims could be supported by more detailed analysis. Providing concrete timing metrics (e.g., runtime comparisons, memory benchmarks across different hardware) would bolster the argument for LFM's practical benefits. More information can be found in Questions part.**
> >
> > We appreciate this question and provide some clarifications. We first refer to answers to the previous question regarding the specification of training batches and model sizes. To demonstrate faster convergence and reduced memory usage, we empirically verified that:
> > - Under the same number of training epochs and identical model size, LFM achieves superior generation performance compared to a global flow (e.g., InterFlow or FM). We acknowledge that in terms of absolute runtime, training an LFM block with $1/K$ model size relative to the global flow might take more than $1/K$ total runtime. This discrepancy can occur due to implementation efficiency and the non-linear relationship between training time and model size. For example, when all else equal, the training of a 464M parameter global flow on a 128x128 Flowers dataset took 2.5~2.75x longer than the training of a 116M parameter local flow on a single A100 GPU.
> > - Due to the smaller size of each LFM block compared to the global flow, training individual LFM blocks consumes less memory than the global flow. For instance, when training the 464M parameter global flow on 128x128 flow data, the model can only utilize a batch size of 40 on a single A100 GPU. In contrast, the 1/4 smaller model with 116M parameters could accommodate a batch size of 130 on the same device.

---

> ### Comment · Reviewer_vDPz · 2024-12-01
>
> Thank you for your detailed rebuttal. I am generally satisfied with your responses to Questions 2, 3, and 5. Below, I provide feedback on other points:
>
> **Regarding Question 1**
> It would be beneficial to include additional comparisons to related works, such as *Improving and Generalizing Flow-Based Generative Models with Minibatch Optimal Transport* (Tong et al., 2023) and *Simulation-free Schrödinger Bridges via Score and Flow Matching*, which similarly focus on straightening vector fields to improve transport efficiency. Additionally, contrasting the trajectory visualizations of your method with FM models, such as the right part of Figure 1 in Tong et al. (2023), would provide more clarity and context.
>
> **Regarding Question 4**
> Thank you for directing me to the theoretical analysis in Section 4. While the assumptions in Proposition 4.1 and Theorem 4.2 are mathematically reasonable, I have concerns about their alignment with the experimental setting. For instance, in high-dimensional tasks like image generation, real-world data distributions are often multimodal or highly non-Gaussian, which may deviate from the theoretical assumptions. Further discussion or validation of these assumptions in such practical contexts would strengthen the paper.
>
> **Regarding Questions 6, 7**
> Thank you for the additional experimental results provided in the official comment. However, I did not find results that conclusively demonstrate the training efficiency advantage of your method. Specifically:
> 1. Your comparison uses a training scheme with advantages for LFM (e.g., batch size and number of training batches) against FM. This is not a fair comparison unless FM is also tested under optimal conditions or a comparable training scheme. Including FM results with its best (or near-best) FID scores would provide a more balanced comparison.
> 2. The rebuttal mentions the non-linear relationship between training time and model size, which is a valid point. However, direct experimental comparisons are necessary to substantiate LFM's training efficiency. For example, with same model size, reporting either the runtime required to achieve comparable FID scores or the FID scores achieved within comparable runtimes would better demonstrate the claimed efficiency advantages.
>
> These points are not addressed in the rebuttal. Therefore, I maintain my score of 3.

---

> > ### Author Response · Authors · 2024-12-02
> >
> > Thank you for your follow-up questions and we answer them below one by one.
> >
> > 1. **It would be beneficial to include additional comparisons to related works, such as Improving and Generalizing Flow-Based Generative Models with Minibatch Optimal Transport (Tong et al., 2023) and Simulation-free Schrödinger Bridges via Score and Flow Matching, which similarly focus on straightening vector fields to improve transport efficiency. Additionally, contrasting the trajectory visualizations of your method with FM models, such as the right part of Figure 1 in Tong et al. (2023), would provide more clarity and context.**
> >
> >     ---
> >     We additionally compared our LFM with FM (Lipman et al., 2023), OTCFM (Tong et al., 2023a), and SBCFM (Tong et al., 2023b). Since LFM is compatible with any interpolation scheme, we use these FM models to train sub-flows within LFM. On the 2D example from the former response to this question, where $P=\mathcal{N}(\mu, I_2)$ with $\mu=[-0.05, -0.05]$, we compared trajectories starting from points sampled from $P$. The results (see [figure](https://anonymous.4open.science/r/LFM_ICLR-B14C/trajectory_together.pdf)) show that our local versions produce trajectories comparable to FM models, with reduced trajectory lengths in some cases (e.g., against FM in the first column). Future work will include quantitative comparisons on larger-scale experiments, particularly against OTCFM and SBCFM.
> >
> >     Ref:
> >     - (Lipman et al., 2023) Flow Matching for Generative Modeling
> >    - (Tong et al., 2023a) Improving and Generalizing Flow-Based Generative Models with Minibatch Optimal Transport
> >    - (Tong et al., 2023b) Simulation-free Schrödinger Bridges via Score and Flow Matching
> >
> > 2. **Thank you for directing me to the theoretical analysis in Section 4. While the assumptions in Proposition 4.1 and Theorem 4.2 are mathematically reasonable, I have concerns about their alignment with the experimental setting. For instance, in high-dimensional tasks like image generation, real-world data distributions are often multimodal or highly non-Gaussian, which may deviate from the theoretical assumptions. Further discussion or validation of these assumptions in such practical contexts would strengthen the paper.**
> >
> >     ---
> >     When the data distribution $P$ lacks a density or does not meet regularity conditions (e.g., in image generation), Corollary 4.3 provides a similar guarantee after applying short-time diffusion, where $p_0 = p_{\delta} = (\rm{OU})^{\delta}_0(P)$. In practice, this corresponds to adding a small amount of noise to $P$, a common dequantization technique for handling such distributions.
> >
> > 3. **Your comparison uses a training scheme with advantages for LFM (e.g., batch size and number of training batches) against FM. This is not a fair comparison unless FM is also tested under optimal conditions or a comparable training scheme. Including FM results with its best (or near-best) FID scores would provide a more balanced comparison.**
> >
> >     ---
> >     The hyperparameter choices (e.g., batch size and training batches) were not specifically optimized for LFM. In fact, our setup favors FM, as LFM trains smaller sub-flows, enabling larger batch sizes if needed (e.g., 130 rather than 40 on a single A100 GPU for the Flower128 dataset). For FM, training a single large model limits the batch size due to GPU memory constraints (e.g., 40 with the same hardware).
> >     Across all experiments, the training schemes for LFM and FM were comparable, with identical batch size, training batches, network parameterization, total parameters, and optimizer. In future work, we will further optimize both methods (e.g., longer training or larger batch sizes for LFM) as suggested.
> >
> > 4. **The rebuttal mentions the non-linear relationship between training time and model size, which is a valid point. However, direct experimental comparisons are necessary to substantiate LFM's training efficiency. For example, with same model size, reporting either the runtime required to achieve comparable FID scores or the FID scores achieved within comparable runtimes would better demonstrate the claimed efficiency advantages.**
> >
> >     ---
> >     Since we compared LFM and FM under the same number of training batches, we interpret “comparable runtimes” as aligning absolute wall-clock training time. To address this, we reran LFM on Flowers128 and CIFAR10 with 2.5× fewer batches than FM, resulting in nearly identical wall-clock times. As shown in the table below, LFM achieves comparable performance on CIFAR10 and improved FID scores on the 128×128 Flowers128 dataset. With improved implementation (e.g., optimizing neural network efficiency for varying model sizes), LFM could be trained for more batches without significantly increasing wall-clock time.
> >     | | LFM | FM|
> >     |-|-|-|
> >     |Flowers128|**65.0**|70.8|
> >     |CIFAR10|12.34|12.30|
> >
> >     Table: FID with nearly identical wall-clock training time on the image generation datasets.

---

### Official Review · Reviewer_K69V · 2024-10-27

**Soundness:** 2
**Presentation:** 3
**Contribution:** 2
**Rating:** 3
**Confidence:** 3

**Summary:**

This paper presents Local Flow Matching (LFM), an enhancement of Flow Matching (FM) for faster, efficient generative modeling. LFM divides the process into sequential FM sub-models, each bridging distributions that are progressively closer from data to noise, allowing smaller models and quicker training. This stepwise approach also enables the use of distillation techniques to further accelerate data generation. Theoretically, LFM provides a generation guarantee based on $\chi^2$-divergence. Experiments show LFM’s improved training efficiency and strong performance in generating tabular data, images, and robotic manipulation policies.

**Strengths:**

The paper explores integrating flow-matching submodels into diffusion processes to enable faster and more efficient learning and inference. The approach is novel and is supported by theoretical guarantees of generation. Experimentally, the method can be applied to various tasks, including image generation, tabular data generation, and robotic manipulation.

**Weaknesses:**

1. More details should be provided, such as the number of function evaluations (NFEs) and the used ODE sampler for all methods in Table 1 and Table 2, to better demonstrate LFM's effectiveness.

2. Although the tasks are diverse, the paper lacks a solid comparison with prior methods on some fundamental tasks. For instance, comparing LFM with Rectified Flow and OU diffusion on CIFAR-10 with the same amount of batches would offer a clearer understanding of LFM's training efficiency and distillation effects.

3. The advantage of LFM on fast training has not been demonstrated well except Table 2.

**Questions:**

1. Can you clarify more on how the method benefits fast training based on the experimental results?

---

> ### Author Response · Authors · 2024-11-25
> **Response to Reviewer K69V**
>
> Please see our first question in the common response regarding the comparison of LFM with FM, where the faster training of LFM is demonstrated by the fact that under identical model sizes, LFM reaches smaller FID than the baseline methods after we train them for the same number of batches.
>
> 1. **(Weakness) More details should be provided, such as the number of function evaluations (NFEs) and the used ODE sampler for all methods in Table 1 and Table 2, to better demonstrate LFM's effectiveness.**
>
> We employ the Dormand-Prince-Shampine (dopri5) ODE sampler for LFM, with tolerances of 1e-5 for non-image examples (Table 1) and 1e-4 for 32x32 images (Table 2). NFE information for most methods in Table 1 is unavailable. However, we compare the NFE and FID performance of LFM against InterFlow and FM below, where we note that
> - LFM achieves lower FID than both methods.
> - While LFM's total NFEs across all local blocks exceed that of the global flow, each NFE uses a smaller model (1/4 size of global flow’s because we used 4 blocks).
> - In the last column, adjusting the $\{ \gamma_n \}$ timestep schedule significantly reduces LFM's NFE with small impact on the FID (still better than FM’s FID).
>
> Table: FID and NFE comparison of LFM vs. InterFlow/FM on Flowers 128x128, maintaining equal total model parameters (in millions) across all four LFM blocks and the global flow. All methods trained for 40K batches (batch size 40), using dopri5 ODE sampler with 1e-3 relative & absolute tolerance.
>
> ||LFM (Trig interpolant)|InterFlow|LFM (OT interpolant)|Flow Matching|LFM (OT interpolant) with adjusted $\{\gamma_n\}$ schedule|
> |-|-|-|-|-|-|
> |FID|59.7|65.9|55.7|70.8|59.7|
> |(NFE, model size per function evaluation)|(182, 116)|(81, 463)|(138, 116)|(66, 463)|(84, 116)|
>
>
> 2. **(Weakness) Although the tasks are diverse, the paper lacks a solid comparison with prior methods on some fundamental tasks. For instance, comparing LFM with Rectified Flow and OU diffusion on CIFAR-10 with the same amount of batches would offer a clearer understanding of LFM's training efficiency and distillation effects.**
>
> We refer the reviewer to the first question in the common response regarding the comparison of LFM with FM and InterFlow on CIFAR-10, Imagenet-32, and Flowers-128, where we demonstrated both faster training efficiency and improved distillation effects.
>
> To further demonstrate the advantage on distillation, we additionally compare LFM with $K$-Rectified Flow (Liu et al., 2023) for $K=2, 3$ (1-Rectified Flow = FM). The Table below shows that LFM reaches better distillation effects than both FM and $K$-Rectified Flow for $K>1$.
>
> Table: FID comparison of LFM and FM & $K$-Rectified Flow before and after distillation (into 4 NFEs) on Flowers 128x128. Note that the training data for the $K$-Rectified Flow is generated by the $(K-1)$-Rectified Flow based on (Liu et al., 2023), so that pre-distillation FID is worse as $K$ increases.
> ||LFM|FM (=1-Rectified Flow)|2-Rectified Flow|3-Rectified Flow|
> |-|-|-|-|-|
> |Pre-distillation|55.7|55.7|62.3|65.4|
> |Distilled @ 4 NFEs|**76.9**|84.6|82.8|82.7|
>
> Ref:
>
> Liu et al., 2023: Flow Straight and Fast: Learning to Generate and Transfer Data with Rectified Flow. ICLR 2023

---

### Official Review · Reviewer_9UHe · 2024-11-04

**Soundness:** 3
**Presentation:** 3
**Contribution:** 3
**Rating:** 5
**Confidence:** 3

**Summary:**

This paper proposes to divide the original Flow Matching (FM) into a sequence of parts, called Local Flow Matching (LFM). The authors provide theoretical analysis of the convergence of the forward process, and the theoretical guarantee of the $\chi^2$ divergence between the real data and generated data distributions. The authors evaluated the proposed method on different tasks, tabular data generation, image generation, robotic manipulation and so on.

**Strengths:**

The authors proposed the Local Flow Matching (LFM), which divides the Flow Matching into a sequence of local parts. The approach is intuitively correct. The authors provided theoretical analysis of the convergence of the forward process, showing that the $\chi^2$ divergence between the noise distribution and noised real data distribution decreases exponentially w.r.t. the number of sub-flows. Furthermore, the authors presented the theoretical results showing the $\chi^2$ divergence between the real distribution and the generated distribution is close w.r.t. the flow training loss $\epsilon$.

**Weaknesses:**

From the experiments, it turns out that the proposed LFM is not working as well as the original Flow Matching (FM) (Lipman et al., 2023). The proposed method achieved an FID of 8.45 on the CIFAR-10 dataset whereas the FM achieved a lower FID of 6.35 on the same dataset. On ImageNet 32x32 dataset, the proposed method achieved an FID of 7.0, but the FM achieved a lower FID of 5.02. The authors should compare with FM on the CIFAR-10, ImageNet 32x32, the Oxford Flowers and the LSUN Church datasets.

The proposed LFM seems to require more training time than the original FM because for training $v_n$ we need to inference from $v_1$ to $v_{n-1}$ for each sample.

The proposed method also need more inference time since we need to inference from $v_1$ to $v_N$.

**Questions:**

What does $g \bot x'_l$ mean in Eq. 6? and why do we need this constraint?

---

> ### Author Response · Authors · 2024-11-25
> **Response to Reviewer 9UHe**
>
> Please see our first question in the common response regarding the comparison of LFM with FM.
>
> 1. **(Weakness) The proposed LFM seems to require more training time than the original FM because for training $v_n$ we need to inference from $v_1$ to $v_{n-1}$ for each sample.**
>
> We clarify that training the $n$-th block requires only the empirical push-forward distribution from the $(n-1)$-th block. Crucially, this distribution is computed *only once, before* training of the $n$-th block begins (see line 5, Algorithm 1). Thus, we avoid repeated inference through all $n-1$ blocks while training the $n$-th block.
>
> 2. **(Weakness) The proposed method also need more inference time since we need to inference from**
>
> We note that throughout the entire experiment section, the total number of parameters over all $v_1,\ldots, v_N$ are the same as the global flow for InterFlow or FM. Please see the second question in the common response above regarding this. Thus, having to inference from $v_1$ to $v_N$ does not necessarily introduce additional inference time because each $v_i$ is proportionally smaller.
>
> 3. **(Question) What does $g \perp x’_l$ mean in Eq. 6? and why do we need this constraint?**
>
> This notation means $g$ and $x’_l$ are independently sampled. We need this constraint so that $x_r$ would correctly follow the distribution $p_n^*$.

---

### Author Response · Authors · 2024-11-25
**Common response to all reviewers**

We appreciate the comments and insights from all reviewers. We have revised the paper with revised parts highlighted in blue. Below, we first address common questions and then address individual ones.

1. **Comparison of LFM with FM (Lipman et al., 2023).**

   **Ref:**

   **Lipman et al. 2023: Flow Matching for Generative Modeling. ICLR 2023.**

We emphasize that in the submitted version, LFM already demonstrated superior performance to FM on non-image tasks, showing improved training efficiency with equivalent model size and training epochs:
- Figure 2: LFM achieved better 2D data generation, both qualitatively and quantitatively (lower NLL).
- Table 4: For robotic manipulation, LFM attained higher success rates than FM on Can, Transport, and Toolhang tasks.

To strengthen our argument, we conducted additional comparisons between LFM and FM on CIFAR-10, Imagenet-32, and Flowers-128 image tasks. For fairness, we maintained equal total model parameters across all LFM blocks and FM's global flow. The updated Table 2 is included in the revision and copied below for convenience. We highlight that with identical training batch count and size, LFM achieves lower FID values than FM, indicating faster training efficiency.

Table: FID (lower is better) comparison of LFM against InterFlow and FM under same model sizes. FIDs with the symbol * are quoted from the original publication. Note that InterFlow uses the Trig interpolant and FM uses the OT interpolant to train the global flow.

| | | CIFAR-10| Imagenet-32|Flowers-128|
|-|-|-|-|-|
| LFM (Trig interpolant)| FID| **8.45**        | **7.00**              | **59.7**           |
|                           | Batch size| **200**             | **256**                   | 40                 |
|                           | # of batches| **5 × 10^4**        | **2 × 10^5**              | 4 × 10^4           |
| InterFlow| FID|  10.27*      | 8.49*             | 65.9           |
|                           |Batch size| 400             | 512                   | 40                 |
|                           |# of batches|5 × 10^5        | 6 × 10^5              | 4 × 10^4           |
||||||
| LFM (OT interpolant)|FID| **8.55**        | **7.20**              | **55.7**           |
|                           |Batch size| 200             | 256                   | 40                 |
|                           |# of batches| 5 × 10^4        | 2 × 10^5              | 4 × 10^4           |
|FM      |FID| 12.30|  7.51              | 70.8           |
|                           |Batch size| 200             | 256                   | 40                 |
|                           |# of batches| 5 × 10^4        | 2 × 10^5              | 4 × 10^4           |

Ref:

Albergo & Vanden-Eijnden, 2023: Building normalizing flows with stochastic interpolants. ICLR 2023.


2. **Model size of LFM against global flows. Are they comparable?**

We emphasize that throughout the experiment section, we maintained equal model parameters between LFM and the global flow. Specifically, for a global flow with $M$ parameters and $K$ LFM local flows, each local flow has $M/K$ parameters. The value of $M$ is detailed in Tables A.1 - A.3.

---

### Author Response · Authors · 2024-11-27
**Awaiting rebuttal response**

Dear Reviewers,

We sincerely appreciate your constructive feedback and thoughtful questions, which have greatly enriched our work. We wanted to kindly follow up regarding the response to our rebuttal.

Thank you again for your time and effort in reviewing our submission.

Best regards,
The Authors

---

### Meta-Review · Area_Chair_ciyh · 2024-12-17

**Metareview:**

This paper suggests training a sequence of flow models by breaking the interval of interest to several parts and training each part using Flow Matching to match marginal probabilities defined by an OU process, in particular Variance Preserving (VP).

It is not clear whether breaking the generation problem to a collection of smaller problems (here with VP marginal probability paths) indeed simplifies the problem and/or provides practical benefits, e.g., compared to training a flow directly using this VP marginal probability path over the full interval. The main difference is that the current approach in practice adds data couplings at each interval compared to the global coupling used for the original problem. The experimental part falls short in showing the suggested method has a strong practical benefit in terms of performance, faster training and/or sampling complexity. Furthermore, some implementation details are missing in the experimental part.

**Additional Comments On Reviewer Discussion:**

No additional comments.

---

### Decision · Program_Chairs · 2025-01-22

Reject